# Immature olfactory sensory neurons provide behaviourally relevant sensory input to the olfactory bulb

Jane S. Huang [1], Tenzin Kunkhyen[1], Alexander N. Rangel[1], Taryn R. Brechbill[1], Jordan D. Gregory[1], Emily D. Winson-Bushby [1], Beichen Liu [1,2], Jonathan T. Avon[1], Ryan J. Muggleton [1,2] & Claire E. J. Cheetham[1,2] ✉

Postnatal neurogenesis provides an opportunity to understand how newborn neurons integrate into circuits to restore function. Newborn olfactory sensory neurons (OSNs) wire into highly organized olfactory bulb (OB) circuits throughout life, enabling lifelong plasticity and regeneration. Immature OSNs form functional synapses capable of evoking firing in OB projection neurons but what contribution, if any, they make to odor processing is unknown. Here, we show that immature OSNs provide odor input to the mouse OB, where they form monosynaptic connections with excitatory neurons. Importantly, immature OSNs respond as selectively to odorants as mature OSNs and exhibit graded responses across a wider range of odorant concentrations than mature OSNs, suggesting that immature and mature OSNs provide distinct odor input streams. Furthermore, mice can successfully perform odor detection and discrimination tasks using sensory input from immature OSNs alone. Together, our findings suggest that immature OSNs play a previously unappreciated role in olfactory-guided behavior.

Most mammalian neurons are generated during embryonic or early postnatal life and cannot be replaced if they are damaged or lost. In contrast, newborn neurons in the olfactory system continue to be generated and wire successfully into highly ordered circuits throughout life[1–3]. This endows the olfactory bulb (OB) with a high level of lifelong plasticity[4–9] and enables substantial functional recovery after injury[10–13]. Combined with its optical accessibility, the mouse OB hence provides a unique opportunity to understand how endogenously generated stem cell-derived neurons functionally integrate into established circuits.

One of these postnatally generated populations, the olfactory sensory neurons (OSNs), provides odor input to the OB[14]. In all terrestrial mammals including humans, newborn OSNs are generated throughout life from basal stem cells in the olfactory epithelium (OE)[1,15,16]. OSNs have a finite lifespan: in mice, their half-life is approximately one month, with most surviving for less than three months except under filtered air or pathogen-free conditions[17–19]. OSN neurogenesis is also upregulated following chemical, mechanical or virus-mediated damage to the OE, enabling rapid repopulation even after an almost complete loss of OSNs[10,13,20]. Furthermore, regeneration of sensory input to the OB enables significant functional recovery of odor representations and olfactory-guided behavior within 6–12 weeks[11,12]. Hence, newborn OSNs in both the healthy and the regenerating OE face the challenge of wiring into pre-existing circuits to maintain or restore function, rather than disrupt it. Determining how this is achieved has broad implications both for understanding how functional integration of stem cell-derived neurons can be promoted to repair damage in other brain regions, and for the treatment of anosmia, which can lead to depression and poor nutrition[21,22].

Individual OSNs follow a well-defined developmental pathway. After terminal division of OE basal cells, Ascl1-expressing neuronally-committed intermediate cells briefly become nascent OSNs that

[1]Department of Neurobiology, University of Pittsburgh, 200 Lothrop St, Pittsburgh, PA 15232, USA. [2]Department of Biological Sciences, Carnegie Mellon University, 4400 Fifth Avenue, Pittsburgh, PA 15213, USA. ✉e-mail: cheetham@pitt.edu

express CXCR4 and DBN1 before transitioning to GAP43- and Gγ8-expressing immature OSNs[23–26]. OSNs begin to express olfactory marker protein (OMP) and downregulate GAP43 expression 7–8 days after terminal cell division[27–30]. Mature OSNs express OMP but not GAP43 or Gγ8. Each mature OSN also typically expresses a single odorant receptor (OR) allele[31–35], selected from a repertoire of approximately 1100 functional receptors[36]. The expressed OR determines both the odorant selectivity of the mature OSN[37–39] and the glomerulus to which it projects its axon[38,40,41]. This generates a highly ordered map of odor input to the brain.

In contrast to mature OSNs, whether immature OSNs respond to odorants is unknown. OSNs begin to express ORs just 4 days after terminal cell division, several days before they express OMP[29]. Furthermore, a subset of immature OSNs express mRNA transcripts that encode both ORs and the molecular machinery necessary to transduce odorant binding into action potentials[35]. This suggests that immature OSNs may be capable of odorant binding and signal transduction. We have also shown previously that immature Gγ8-expressing OSNs form synapses with OB neurons in the glomerular layer and that optogenetic stimulation of immature OSN axons evokes robust firing in OB neurons in the glomerular, external plexiform and mitral cell layers[4]. Hence, if immature OSNs can detect and transduce odorant binding, then they may play a previously unknown role in transmitting odor information to OB neurons to support olfactory-guided behavior. However, this would also raise an important question: is input from immature OSNs odorant selective? Recent studies have shown that a subset of immature OSNs express transcripts encoding multiple ORs[35,42]. Therefore, if multi-OR expressing immature OSNs were to provide odor information to the OB, they could degrade the odorant selectivity of glomerular input.

Determining what contribution, if any, immature OSNs make to OB sensory input is essential in understanding both how odor information is processed by OB circuits and how adult born OSNs continue to wire into highly ordered circuits without disrupting existing function. Here, using mice, we employ in vivo 2-photon calcium imaging to demonstrate that immature OSNs provide input to glomeruli that is as odorant selective as that provided by mature OSNs. Furthermore, immature OSNs continue to provide information about concentration differences to glomeruli at high concentrations at which mature OSN responses are already maximal. We also show using optogenetic stimulation that immature OSNs form monosynaptic connections with superficial tufted cells in the OB. Furthermore, we demonstrate that sensory input from immature OSNs is sufficient to mediate odor detection and simple odor discrimination in behavioral tasks.

## Results

### Immature OSNs respond to odorants

To determine whether immature OSNs respond to odorants, we bred Gγ8-tTA;tetO-GCaMP6s (referred to as Gγ8-GCaMP6s) mice, in which immature OSNs selectively express the genetically encoded calcium indicator GCaMP6s (Fig. 1a, b). Unlike recombinase-based expression systems, the tetracycline transactivator system enables transient expression of a reporter protein during a particular developmental stage. We validated the specificity of GCaMP6s expression in immature OSNs in 3-week-old mice, a time point at which large numbers of immature OSNs are present in the OE[43]. Only 3% of Gγ8-GCaMP6s-expressing neurons also expressed OMP (Fig. 1b, c, Fig. S1). Hence, the vast majority of Gγ8-GCaMP6s-expressing OSNs are immature GCaMP6s+OMP- neurons, whereas a very small proportion of GCaMP6s+OMP+ OSNs are in transition to maturity[4].

Both GAP43 staining[44] and expression of Gγ8-tTA-driven reporter proteins[4] have shown that immature OSN axons do enter glomeruli. To confirm that immature OSN axons expressing GCaMP6s innervate glomeruli on the dorsal surface of the OB, which would enable us to image odorant responses in vivo, we next bred OMP-cre;flox-tdT;Gγ8-

tTA;tetO-GCaMP6s (referred to as OMP-tdT-Gγ8-GCaMP6s) mice. This line expresses the red fluorescent protein tdTomato in mature OSN axons and GCaMP6s in immature OSN axons. We implanted acute cranial windows over the OB of four 3-week-old OMP-tdT-Gγ8-GCaMP6s mice and first collected 2-color 2-photon z-stacks of the glomerular layer. While the resting fluorescence of GCaMP6s is low[45], we observed some GCaMP6s-expressing immature OSN axons present within glomeruli, which were defined using OMP-tdT fluorescence (Fig. 1d). We also performed preliminary 2-photon calcium imaging in these mice, using four odorants that are known to activate dorsal OB glomeruli. In some glomeruli, we observed strong odorant-evoked increases in GCaMP6s fluorescence in immature OSN axons (Fig. 1e). Hence, we concluded that immature GCaMP6s-expressing OSN axons do enter glomeruli and can respond to odors.

To quantify odorant responses, we next implanted acute cranial windows over the OB of 3-week-old Gγ8-GCaMP6s and OMP-GCaMP6s mice, in which GCaMP6s is expressed in mature OSNs. We then performed 2-photon calcium imaging of glomeruli innervated by immature or mature OSN axons in response to stimulation with a panel of seven dorsal OB-activating odorants. All responses were blank stimulus-subtracted to account for any potential mechanosensory contribution to evoked responses as a result of changes in air flow (Fig. 2a)[46–50].

Mature OSN axons are present throughout the axodendritic domains of glomeruli and can be used to delineate glomerular borders (Fig. 1d). In contrast, the extent to which individual glomeruli are innervated by immature OSN axons varies, and the density of immature OSN axons may be higher proximal to the olfactory nerve layer and around the periphery of the glomerulus[4,44]. This was also evident for immature OSNs expressing GCaMP6s (Figs. 1d, 2a). Because this could affect our ability to define the borders of individual glomeruli, we did not compare response amplitudes between Gγ8-GCaMP6s and OMP-GCaMP6s mice. It was also important to determine whether differences in glomerular innervation might affect our ability to detect glomeruli in 3-week-old Gγ8-GCaMP6s mice, in which there is a high density of immature OSNs[43]. We quantified the density of detectable glomeruli in 2-photon images from Gγ8-GCaMP6s and OMP-GCaMP6s mice, including glomeruli that were identifiable either from resting fluorescence or following odorant stimulation. We also determined glomerular density in in vivo 2-photon images of the dorsal OB surface of Gγ8-sypGFP-tdT and OMP-sypGFP-tdT mice, which express cytosolic tdTomato and GFP-tagged synaptophysin in immature and mature OSNs respectively[4]. We found no effect of OSN maturity or the reporter protein(s) expressed (GCaMP6s vs. sypGFP-tdT) on glomerular density (Fig. 2b). Hence, we concluded that the relative sparsity of immature vs. mature OSN axons entering glomeruli does not preclude glomerular detection in 3-week-old Gγ8-GCaMP6s mice.

We found that immature OSN axons in many glomeruli responded to at least one odorant in the panel (Fig. 2c). The proportion of glomeruli that responded to each odorant in Gγ8-GCaMP6s mice was similar to that in OMP-GCaMP6s mice (Fig. 2c). There was no significant difference in mean glomerular response amplitude per Gγ8-GCaMP6s mouse across the odorant panel that we tested, although there was a trend for glomerular responses to some odorants to be larger (Fig. 2d). A similar trend that also did not reach statistical significance was seen in OMP-GCaMP6s mice (Fig. 2e).

Modeling has suggested that ephaptic transmission may occur between OSN axons in the olfactory nerve layer[51], making it important to confirm that immature OSNs in the OE do indeed detect odorants. Therefore, we also imaged odorant responses in immature OSNs in the OE of Gγ8-GCaMP6s mice using an ex vivo hemi-head preparation (Fig. 2f). We found that a subset of immature OSNs responded to each of the four odorants that we tested (Fig. 2g), and there was no difference in response amplitudes across odorants (Fig. 2h). Taken together, the data in Fig. 2 provide strong evidence that odorants evoke calcium

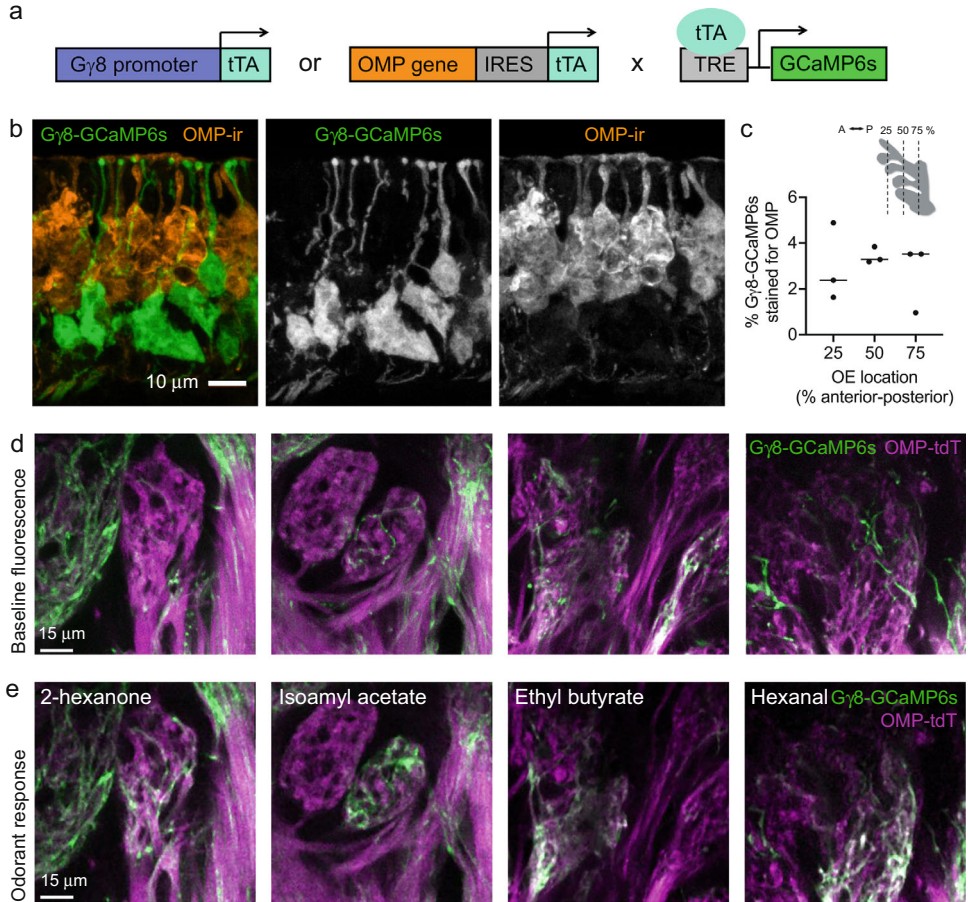

**Fig. 1 | Gγ8-GCaMP6s mice enable selective imaging of immature OSN axon odorant responses in glomeruli. a** Schematic of breeding strategy to generate mice expressing GCaMP6s in either immature or mature OSNs under control of the Gγ8 or OMP promoter, respectively, using the tetracycline transactivator system. **b** MIP of confocal z-stack of coronal OE section from a Gγ8-GCaMP6s mouse showing lack of colocalization with OMP-stained OSNs. **c** Co-expression of OMP (OMP-ir) in Gγ8-GCaMP6s-expressing OSNs in the septal olfactory epithelium (OE) along the anterior-posterior (A-P) axis. Only $3.0 \pm 1.2\%$ of Gγ8-GCaMP6s-expressing neurons co-express OMP (mean ± s.d., $n = 4118$ Gγ8-GCaMP6s-expressing OSNs from 3 mice). Lines: median, symbols: values for individual mice. Inset shows location of analyzed OE sections. **d** In vivo 2-photon images of single optical sections showing baseline fluorescence of glomeruli in OMP-tdT-Gγ8-GCaMP6s mice. **e** Single time point from time series showing peak odorant response for the same optical section as in d. Odorant responses are for stimulation with 2-hexanone, isoamyl acetate, ethyl butyrate and hexanal. **d, e** Glomeruli from four different mice are shown. Source data are provided as a Source Data file.

responses in immature OSN axon terminals, suggesting that immature OSNs can detect and transduce odorant binding.

Recent studies have suggested that mature OSNs can respond to mechanical stimuli using the same signal transduction machinery as for odor input[46–50]. We found that a deodorized air puff stimulus delivered to the external nares elicited putative mechanosensory responses in a similar proportion of glomeruli in Gγ8-GCaMP6s and OMP-GCaMP6s mice (Fig. S2a, b). Both the proportion of responding glomeruli (Fig. S2b) and the amplitude of air puff-induced glomerular responses in Gγ8-GCaMP6s and OMP-GCaMP6s mice (Fig. S2c) were similar to a previous report of mechanosensory responses induced by increased air flow[48]. Together, these data provide evidence for putative mechanosensory responses in immature OSNs (Supplementary Discussion).

### Concentration coding differs with OSN maturity

We next analyzed the effect of odorant concentration on GCaMP6s responses in immature and mature OSNs using four concentrations each of five different odorants (Fig. 3a, b). Previous studies have shown that increased odorant concentration results in the activation of additional glomeruli[52–55]. Consistent with these findings, OMP-GCaMP6s mice exhibited an increase in the percentage of glomeruli responding to each odorant in our panel as odorant concentration was

increased from 0.5% to 10% (Fig. 3c). Similarly, the percentage of glomeruli that responded to each odorant also increased across the 0.5–10% odorant concentration range in Gγ8-GCaMP6s mice (Fig. 3d). To compare generalized effects of odorant concentration across genotypes, we analyzed mouse-odorant pairs. There was no difference in the percentage of glomeruli that responded to each odorant concentration between OMP-GCaMP6s and Gγ8-GCaMP6s mice (Fig. 3e). This suggests that both immature and mature OSN axons innervating additional glomeruli are recruited as odorant concentration increases.

We also compared the amplitudes of responses evoked by 0.5–10% concentrations of the five odorants. Previous studies have found that higher odorant concentrations evoke larger amplitude glomerular responses[52–55]. We also found that mean odorant response amplitude increased with odorant concentration in both OMP-GCaMP6s mice (Fig. 3f) and Gγ8-GCaMP6s mice (Fig. 3g). However, whereas response amplitudes increased sharply between 0.5% and 1% odorant concentrations but then reached a plateau in OMP-GCaMP6s mice, the relationship between response amplitude and odorant concentration appeared shallower and more uniform across the concentration range in Gγ8-GCaMP6s mice. We first confirmed that there was no indication of GCaMP6s saturation even for the largest amplitude responses in OMP-GCaMP6s mice (Fig. S3). We then used mouse-odorant pairs to compare response amplitude at different

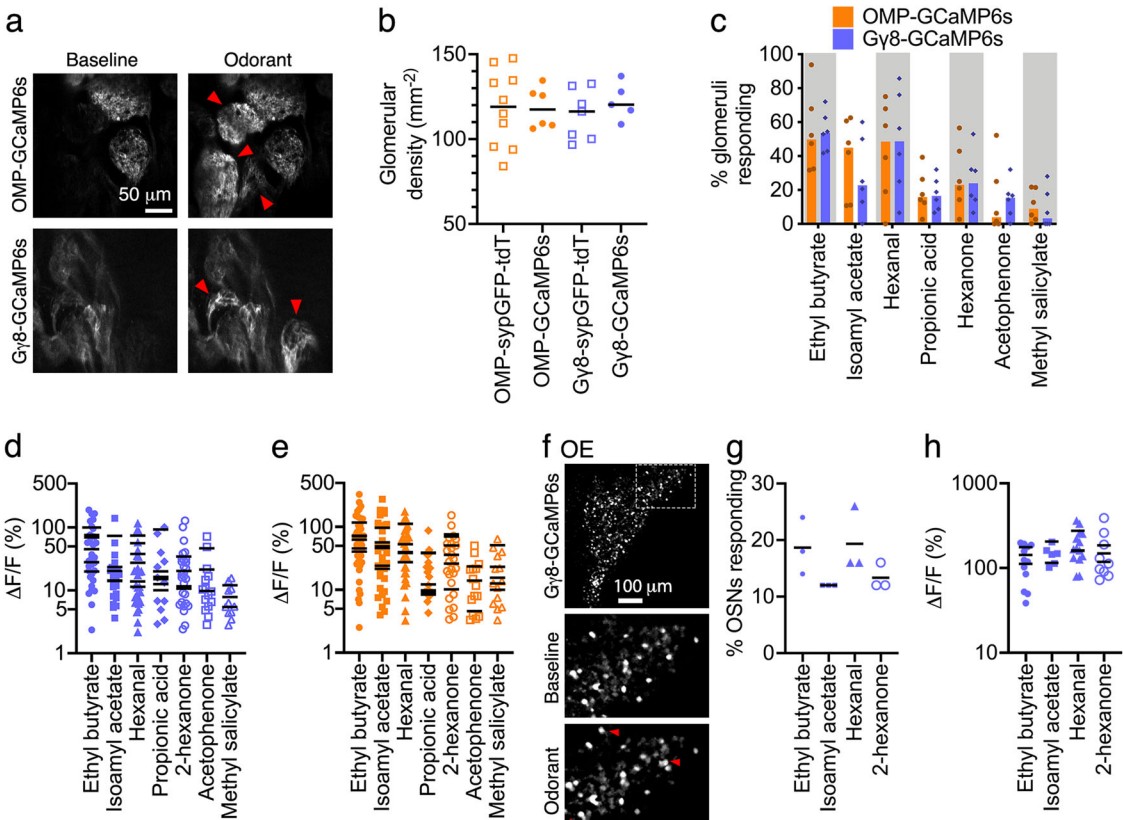

**Fig. 2 | Immature OSNs respond to odorants. a** Example single z-plane 2-photon images during single trial showing baseline fluorescence and odorant-evoked responses in OMP-GCaMP6s and Gγ8-GCaMP6s mice. Red arrows: ethyl butyrate-responsive glomeruli. **b** No effect of OSN maturity or reporter protein(s) on the density of detected glomeruli (Two-way ANOVA. Effect of OSN maturity: $P = 0.99$, $F_{1,24} = 2.93 \times 10^{-5}$. Effect of reporter protein(s): $P = 0.55$, $F_{1,24} = 0.37$. Interaction: $P = 0.57$, $F_{1,24} = 0.33$. $n = 10$ OMP-sypGFP-tdT, 6 OMP-GCaMP6s, 7 Gγ8-sypGFP-tdT and 5 Gγ8-GCaMP6s mice). Lines: mean, symbols: individual mice. **c** A similar percentage of glomeruli respond to each of the 7 odorants in OMP-GCaMP6s vs. Gγ8-GCaMP6s mice (Wilcoxon signed rank test. $P = 0.47$, $W = -10$, $n = 6$ mice per group). Colored bars: median, symbols: individual mice. Gray shaded areas are to aid visualization only. **d** No significant difference in response amplitude between tested odorants in Gγ8-GCaMP6s mice (Nested one-way ANOVA. $P = 0.080$, $F_{6,30} = 2.12$, $n = 6$ mice). **e** No significant difference in response amplitude between

tested odorants in OMP-GCaMP6s mice (Nested one-way ANOVA. $P = 0.073$, $F_{6,31} = 2.17$. $n = 6$ mice). **d, e** Lines: mean per mouse, symbols: individual glomeruli. Not all mice had glomeruli that responded to all odorants. **f** Example single z-plane 2-photon images showing immature OSNs in the OE responding to ethyl butyrate stimulation ex vivo. Top: imaged field of view showing medial OE surface of a Gγ8-GCaMP6s mouse in the hemi-head preparation. Lower: baseline and odorant-evoked fluorescence in boxed region in top image. Red arrows: ethyl butyrate-responsive immature OSNs. **g** Similar percentage of immature OSNs responded to each odorant (Friedman test. $P = 0.15$, Friedman statistic $= 5.44$, $n = 3$ mice). Lines: mean, symbols: individual mice. **h** Immature OSN response amplitudes were similar across odorants (Nested one-way ANOVA. $P = 0.53$, $F_{3,91} = 0.75$, $n = 6–13$ responding OSNs per mouse for each odorant). Lines: mean per mouse, symbols: individual OSNs. All relevant statistical tests were two-tailed. Source data are provided as a Source Data file.

concentrations. In OMP-GCaMP6s mice, response amplitude increased significantly between 0.5% and 1% concentrations, but was not significantly different between 1% and 5%, or 5% and 10% concentrations (Fig. 4a). In contrast, in Gγ8-GCaMP6s mice, there was a significant increase in response amplitude for all three steps in the concentration range (Fig. 4b). We obtained similar results when we instead analyzed glomerulus-odorant pairs. In OMP-GCaMP6s mice, response amplitude increased significantly between 0.5% and 1%, and between 1% and 5%, concentrations, but not between 5% and 10% concentrations (Fig. S4a). In contrast, in Gγ8-GCaMP6s mice, there was again a significant increase in response amplitude for all three steps in the concentration range (Fig. S4b). To confirm that our findings did not arise from mean values being dominated by strongly responding glomeruli, we normalized the response amplitudes for each glomerulus to the response to the 10 % concentration of each odorant, before calculating the mean value per mouse. This ensured that each odorant-responsive glomerulus made an equal contribution to the mean value for that mouse. Comparing mouse-odorant pairs, we again found a similar pattern of results: there was a significant difference in normalized response amplitude between 5% and 10% odorant concentrations in Gγ8-GCaMP6s mice but not in OMP-GCaMP6s mice (Fig. S5a, b).

We also considered that odorant response duration, as well as amplitude, could encode information about odorant concentration. While GCaMP6s does not provide sufficient temporal resolution to resolve individual spikes within a burst[45], we reasoned that the odorant response integral could provide a useful correlate. Comparing mouse-odorant pairs at different concentrations, in OMP-GCaMP6s mice, we found a significant increase in response integral between 0.5% and 1%, and between 1% and 5%, concentrations, but not between 5% and 10% concentrations (Fig. S6a). In contrast, there was a significant increase in response integral between each ascending pair of concentrations in Gγ8-GCaMP6s mice (Fig. S6b). Hence, there is good agreement between our analyses of response amplitude and response integral.

We also determined whether differences in response latency or time to peak provided information about differences in odorant concentration by comparing mouse-odorant pairs. In OMP-GCaMP6s mice, we found that response latency was significantly shorter for 1% odorants vs. 0.5% odorants but was similar across the 1–10% concentration range (Fig. 4c). In contrast, in Gγ8-GCaMP6s mice, response latency decreased with each step across the entire 0.5–10% concentration range (Fig. 4d). Time to peak was similar across odorant concentrations in OMP-GCaMP6s mice (Fig. 4e) and was different only

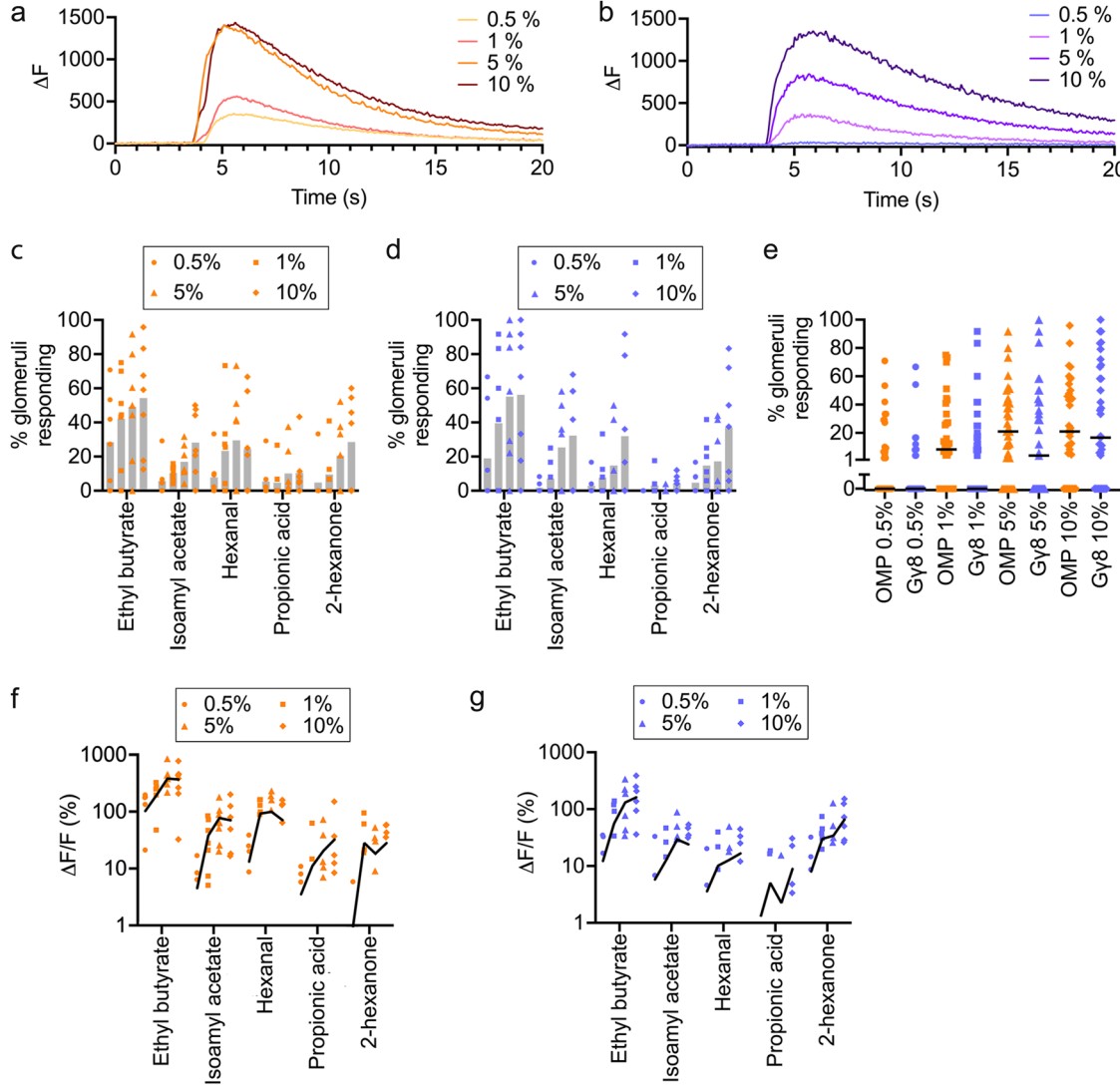

**Fig. 3 | Immature OSNs encode odorant concentration. a** Responses of an example glomerulus from an OMP-GCaMP6s mouse to the four tested concentrations of ethyl butyrate. **b** Responses of an example glomerulus from a Gγ8-GCaMP6s mouse to the four tested concentrations of ethyl butyrate. **c** Percentage of glomeruli responding increases with odorant concentration in OMP-GCaMP6s mice (Two-way ANOVA. Effect of concentration: $P < 0.002$, $F_{3,120} = 5.35$. Effect of odorant: $P < 0.001$, $F_{4,120} = 11.6$. Interaction: $P = 0.98$, $F_{12,120} = 0.36$. $n = 7$ mice per group). Bars: mean, symbols: values for individual mice. **d** Percentage of glomeruli responding increases with odorant concentration in Gγ8-GCaMP6s mice (Two-way ANOVA. Effect of concentration: $P < 0.001$, $F_{3,120} = 8.10$. Effect of odorant: $P < 0.001$, $F_{4,120} = 10.8$. Interaction: $P = 0.78$, $F_{12,120} = 0.66$. $n = 7$ mice per group). Bars: mean, symbols: values for individual mice. **e** Similar percentage of glomeruli respond to each odorant concentration in OMP-GCaMP6s and Gγ8-GCaMP6s mice (One-way ANOVA on Ranks. $P < 0.001$, Kruskal–Wallis statistic = 33.4. Dunn's multiple

comparisons. OMP 0.5% vs. Gγ8 0.5%: $P = 0.97$, $Z = 1.17$. OMP 1% vs. Gγ8 1%: $P = 0.86$, $Z = 1.24$. OMP 5% vs. Gγ8 5%: $P = 0.79$, $Z = 1.29$. OMP 10% vs. Gγ8 10%: $P > 0.99$, $Z = 0.21$. $n = 140$ mouse-odorant pairs per genotype). Lines: median, symbols: mouse-odorant pairs. **f** Mean odorant response amplitude per mouse increases with odorant concentration in OMP-GCaMP6s mice (Two-way repeated measures ANOVA. Effect of concentration: $P < 0.001$, $F_{1.24,37.1} = 14.4$. Effect of odorant: $P < 0.001$, $F_{4,30} = 12.8$. Interaction: $P < 0.001$, $F_{12,90} = 4.45$. $n = 7$ mice). **g** Mean odorant response amplitude per mouse increases with odorant concentration in Gγ8-GCaMP6s mice (Two-way repeated measures ANOVA. Effect of concentration: $P < 0.001$, $F_{1.27,38.1} = 18.1$. Effect of odorant: $P = 0.004$, $F_{4,30} = 4.89$. Interaction: $P < 0.001$, $F_{12,90} = 5.85$. $n = 7$ mice). **f, g** Lines: connect mean values for each odorant, symbols: values for individual mice. All statistical tests were two-tailed. Source data are provided as a Source Data file.

between 5% and 10% concentrations in Gγ8-GCaMP6s mice (Fig. 4f). Overall, we concluded that for the odorants and concentration range tested here, both immature and mature OSNs encode concentration, but immature OSNs provide information about differences between higher concentrations that is not available from mature OSN input.

### Immature OSNs respond selectively to odorants

Some immature OSNs express mRNA transcripts encoding multiple ORs[35,42]. In particular, one study showed that some late immature OSNs, which already express transcripts encoding all of the essential olfactory signal transduction machinery, still express multiple OR

transcripts[35]. If this is also the case at the protein level, immature OSNs innervating each glomerulus may respond to a wider range of odorants. Therefore, we compared the selectivity of odorant responses in individual glomeruli in Gγ8-GCaMP6s and OMP-GCaMP6s mice. We found that the number of odorants to which an individual glomerulus responded was similar in OMP-GCaMP6s and Gγ8-GCaMP6s mice (Fig. 5a). However, it is possible that selectivity is also reflected in response magnitude, with mature OSNs responding strongly to a small number of odorants and weakly to others, whereas immature OSN axons innervating glomeruli may respond more similarly to multiple odorants. Therefore, we also evaluated odorant selectivity by

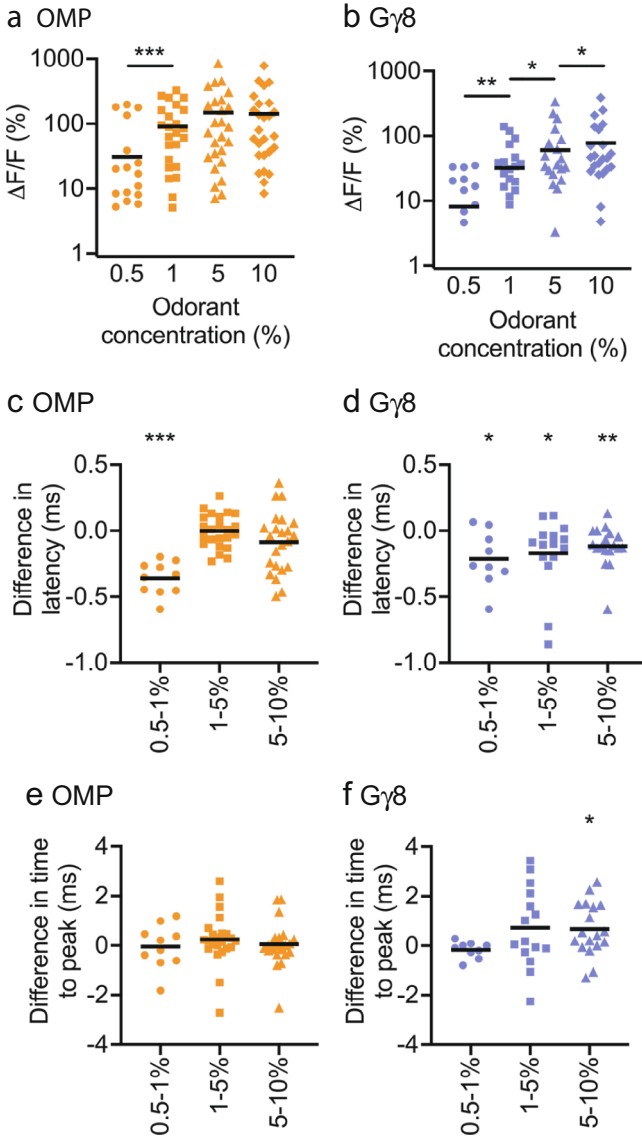

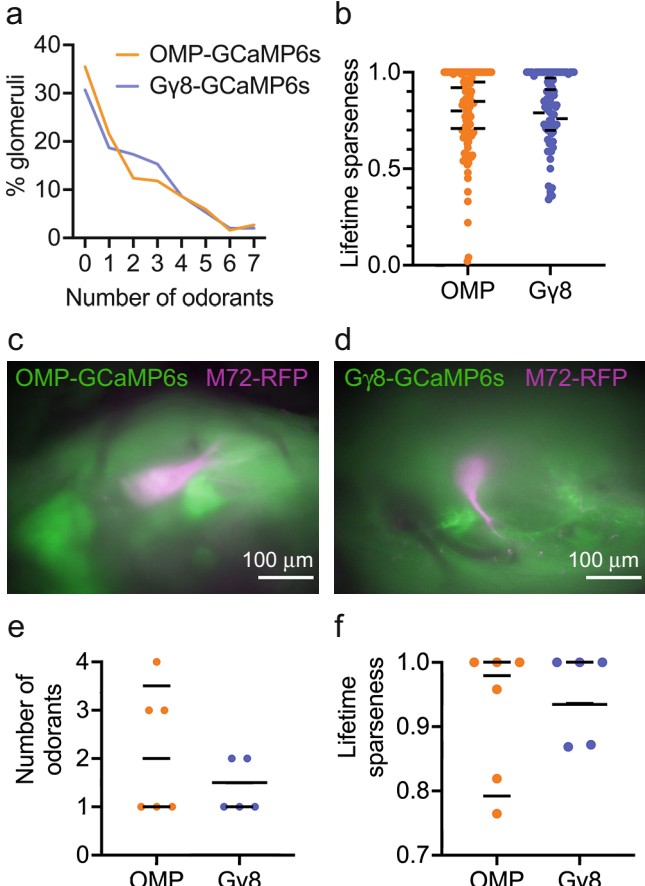

**Fig. 5 | Immature and mature OSN axons innervating the M72 glomerulus show similar odorant selectivity. a** Glomeruli in OMP-GCaMP6s and Gγ8-GCaMP6s mice respond to a similar number of odorants in a seven-odorant panel (Wilcoxon signed rank test. $P = 0.95$, $W = -2$, $n = 186$ glomeruli in 6 OMP-GCaMP6s mice and 150 glomeruli in 6 Gγ8-GCaMP6s mice). **b** Similar lifetime sparseness of glomerular odorant responses in OMP-GCaMP6s and Gγ8-GCaMP6s mice (Nested $t$-test. $P = 0.97$, $t = 0.037$. $n = 120$ responding glomeruli in 6 OMP-GCaMP6s mice and 97 responding glomeruli in 6 Gγ8-GCaMP6s mice). Lines: mean values for each mouse, symbols: individual glomeruli. **c** Camera image showing M72 glomerulus (magenta) and baseline GCaMP6s fluorescence (green) in mature OSN axons in vivo through a cranial window. **d** Camera image showing M72 glomerulus (magenta) and baseline GCaMP6s fluorescence (green) in immature OSN axons in vivo through a cranial window. **e** Mature and immature OSN axons in lateral M72 glomeruli respond to a similar number of odorants (Nested $t$-test. $P = 0.36$, $t = 1.03$, $n = 6$ glomeruli from 3 OMP-GCaMP6s-M72-RFP mice and 5 glomeruli from 3 Gγ8-GCaMP6s-M72-RFP mice). Lines: mean for each mouse, symbols: individual M72 glomeruli. **f** Lifetime sparseness of odorant responses of mature and immature OSN axons in lateral M72 glomeruli is similar (Nested $t$-test. $P = 0.70$, $t = 0.42$, $n = 6$ glomeruli from 3 OMP-GCaMP6s-M72-RFP mice and 5 glomeruli from 3 Gγ8-GCaMP6s-M72-RFP mice). Bars: mean for each mouse, symbols: individual M72 glomeruli. Long bar in Gγ8 group represents two mice. All statistical tests were two-tailed. Source data are provided as a Source Data file.

**Fig. 4 | Immature OSNs provide information about differences between high odorant concentrations. a** Odorant response amplitudes increase significantly between 0.5–1% concentrations but are similar across higher concentrations for OMP-GCaMP6s mouse-odorant pairs (One-way repeated measures ANOVA. $P = 0.001$, $F_{1,19,32} = 11.0$. Sidak's multiple comparisons. 0.5% vs. 1%: $P < 0.001$, $t = 5.47$. 1% vs. 5%: $P = 0.092$, $t = 2.27$. 5% vs. 10%: $P = 0.80$, $t = 0.83$. $n = 28$ mouse-odorant pairs). **b** Odorant response amplitudes increase significantly between each ascending concentration pair for Gγ8-GCaMP6s mouse-odorant pairs (One-way repeated measures ANOVA. $P < 0.001$, $F_{1,20,27.7} = 13.2$. Sidak's multiple comparisons. 0.5% vs. 1%: $P = 0.002$, $t = 3.91$. 1% vs. 5%: $P = 0.030$, $t = 2.80$. 5% vs. 10%: $P = 0.023$, $t = 2.92$. $n = 28$ mouse-odorant pairs). **c** Response latency is shorter for 1% vs. 0.5 % odorant concentrations but similar between other concentration pairs in OMP-GCaMP6s mice (Bonferroni-corrected one-sample $t$-tests vs. zero. 0.5–1%: $P < 0.001$, $t = 9.03$, $n = 10$. 1–5%: $P = 0.94$, $t = 0.082$, $n = 23$. 5–10%: $P = 0.25$, $t = 0.60$, $n = 23$). **d** Response latency decreases as odorant concentration increases in Gγ8-GCaMP6s mice (Bonferroni-corrected one-sample $t$-tests vs. zero. 0.5–1%: $P = 0.016$, $t = 3.14$, $n = 9$. 1–5%: $P = 0.030$, $t = 2.42$, $n = 15$. 5–10%: $P = 0.005$, $t = 3.24$, $n = 18$). **e** Similar time to peak across odorant concentrations in OMP-GCaMP6s mice (Bonferroni-corrected one-sample $t$-tests vs. zero. 0.5–1%: $P = 0.88$, $t = 0.16$, $n = 10$. 1–5%: $P = 0.27$, $t = 1.12$, $n = 23$. 5–10%: $P = 0.80$, $t = 0.26$, $n = 23$). **f** Time to peak is longer for responses to 10% vs. 5% odorant concentrations but similar across lower concentrations in Gγ8-GCaMP6s mice (Bonferroni-corrected one-sample $t$-tests vs. zero. 0.5–1%: $P = 0.17$, $t = 1.53$, $n = 9$. 1–5%: $P = 0.10$, $t = 1.75$, $n = 15$. 5–10%: $P = 0.017$, $t = 2.65$, $n = 18$). **a–f** Lines: mean, symbols: mouse-odorant pairs. All statistical tests were two-tailed. Source data are provided as a Source Data file.

calculating the lifetime sparseness $(S_L)$[56] of glomerular odorant responses. $S_L$ provides a measure of the breadth of odor tuning for each glomerulus that accounts for the amplitude of odorant-evoked responses: a glomerulus with $S_L = 0$ responds equally to all seven odorants whereas a glomerulus with $S_L = 1$ responds to a single odorant. We found that $S_L$ was similar in OMP-GCaMP6s and Gγ8-GCaMP6s mice, with most glomeruli responding to a small number of odorants in the panel (Fig. 5b).

It is also possible that odorant selectivity varies with glomerular identity. To directly compare the odorant selectivity of immature and

mature OSN axons in the same glomerulus, we bred Gγ8-GCaMP6s and OMP-GCaMP6s mice that were also homozygous for the M72-RFP allele (referred to as Gγ8-GCaMP6s-M72-RFP and OMP-GCaMP6s-M72-RFP mice; Fig. 5c, d). We used the same odorant panel but with the addition of 2-hydroxyacetophenone, a known high affinity M72 receptor ligand[57,58]. As expected, all lateral M72 glomeruli responded to 2-hydroxyacetophenone, and within lateral M72 glomeruli, immature OSN axons did not respond to a larger number of odorants than did mature OSN axons (Fig. 5e). Furthermore, the lifetime sparseness of M72 odorant responses was similar for M72 glomeruli in OMP-GCaMP6s-M72-RFP and Gγ8-GCaMP6s-M72-RFP mice (Fig. 5f). As glomerular position is relatively stereotyped across animals[59–61], we reasoned that by imaging odorant responses of glomeruli in a field of view centered on the lateral M72 glomerulus, we could compare immature and mature OSN odorant responses in a more similar subset of glomeruli than in our original data set. With this approach, we again found no difference in the number of odorants to which glomeruli responded or the lifetime sparseness of odorant responses between OMP-GCaMP6s-M72-RFP and Gγ8-GCaMP6s-M72-RFP mice (Fig. 6a, b).

Finally, we compared the lifetime sparseness of odorant-responsive glomeruli in OMP-GCaMP6s and Gγ8-GCaMP6s mice that

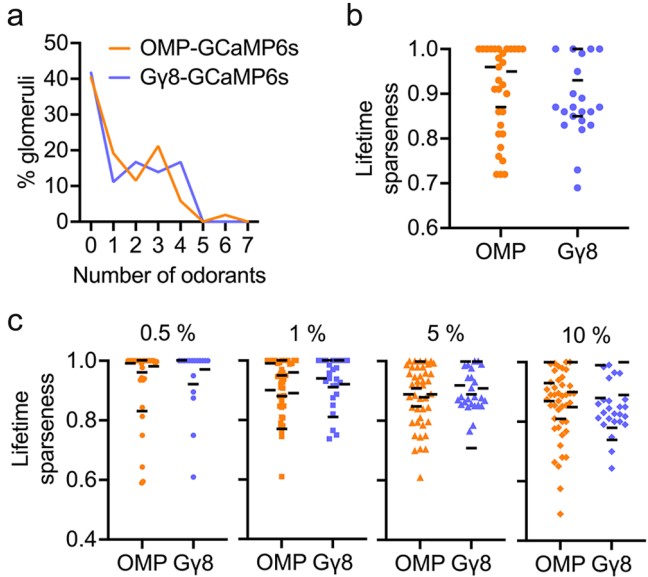

**Fig. 6 | Similar odorant selectivity of immature and mature OSN axons across multiple glomeruli and a range of odorant concentrations. a** Glomeruli surrounding the lateral M72 glomerulus in OMP-GCaMP6s-M72-RFP and Gγ8-GCaMP6s-M72-RFP mice respond to a similar number of odorants in an eight-odorant panel (Wilcoxon signed rank test. P > 0.99, W = −1, n = 52 glomeruli from 3 OMP-GCaMP6s-M72-RFP mice and 36 glomeruli from 3 Gγ8-GCaMP6s-M72-RFP mice). **b** Lifetime sparseness of odorant responses of mature and immature OSN axons in glomeruli surrounding the lateral M72 glomerulus is similar (Nested t-test. P = 0.67, t = 0.45, n = 31 odorant-responsive glomeruli in 3 OMP-GCaMP6s-M72-RFP mice and 21 odorant-responsive glomeruli in 3 Gγ8-GCaMP6s-M72-RFP mice). Lines: mean for each mouse, symbols: individual glomerulus. **c** Lifetime sparseness of odorant responses of mature and immature OSN axons is similar across four different odorant concentrations (Nested t-tests. 0.5%: P = 0.70, t = 0.40, n = 71 odorant-responsive glomeruli from 7 OMP-GCaMP6s mice and 30 odorant-responsive glomeruli from 7 Gγ8-GCaMP6s mice. 1%: P = 0.83, t = 0.22, n = 115 odorant-responsive glomeruli from 7 OMP-GCaMP6s mice and 61 odorant-responsive glomeruli from 7 Gγ8-GCaMP6s mice. 5%: P = 0.95, t = 0.060, n = 77 odor-responsive glomeruli from 7 OMP-GCaMP6s mice and 81 odorant-responsive glomeruli from 7 Gγ8-GCaMP6s mice. 10%: P = 0.55, t = 0.62, n = 138 odorant-responsive glomeruli from 7 OMP-GCaMP6s mice and 85 odorant-responsive glomeruli from 7 Gγ8-GCaMP6s mice). Lines: mean for each mouse, symbols: individual glomerulus. All relevant statistical tests were two-tailed. Source data are provided as a Source Data file.

had been stimulated with four different odorant concentrations (Figs. 3, 4). We found that lifetime sparseness was similar across genotypes for all four odorant concentrations (Fig. 6c). Overall, we concluded that glomerular odorant selectivity was similar for immature and mature OSN axons.

## Immature OSNs form monosynaptic connections with OB neurons

To provide sensory information to the OB, immature OSNs must not only detect and transduce odorant binding but also transmit this information to OB neurons. We have shown previously that optogenetic stimulation of immature OSNs elicits robust stimulus-locked firing in OB neurons in vivo[4]. However, our extracellular recording approach meant that we could not determine whether immature OSNs provide monosynaptic input to OB neurons. To address this question, we performed OB slice electrophysiology using P18–P25 Gγ8-ChIEF-Citrine and OMP-ChIEF-Citrine mice, in which the blue light-activated cation channel ChIEF fused to the yellow fluorescent protein Citrine is expressed selectively in immature and mature OSNs, respectively[4] (Fig. 7a–d). We then made whole-cell recordings of superficial tufted cells (STCs), which are known to receive direct monosynaptic input from mature OSNs[62].

STCs were identified using morphological and physiological criteria: they reside at the border between the glomerular and external plexiform layers, possess a lateral dendrite in the superficial EPL in addition to the primary apical dendrite, and exhibit regular or irregular non-bursting spiking patterns without the depolarizing envelope characteristic of external tufted cells (Fig. 7e, f)[62–68]. We targeted STCs located close to glomeruli innervated by ChIEF-Citrine expressing axons, voltage clamped them at −70 mV, and recorded their responses to multiglomerular optogenetic stimulation (Fig. 7g, h; Fig. S7). Recordings were made with APV in the bath to isolate fast AMPA-mediated currents. We compared responses evoked by 1 ms blue light photoactivation of mature vs. immature OSN axons. A similar proportion of STCs from OMP-ChIEF-Citrine and Gγ8-ChIEF-Citrine mice showed inward currents in response to optogenetic stimulation (Fig. 8a). For a subset of recorded neurons, we confirmed that the responses were AMPA-mediated as they were completely abolished by bath application of NBQX (Fig. 7g, h).

We next compared the properties of the excitatory inputs received by STCs from OMP-ChIEF-Citrine axons vs. Gγ8-ChIEF-Citrine axons (Fig. 8b). There was no significant difference in the peak amplitude of EPSCs evoked by optogenetic stimulation of mature vs. immature OSN axons (Fig. 8c; Fig. S8a). Light-evoked EPSCs recorded in STCs from both genotypes displayed short onset latencies and had low trial-to-trial jitter (Fig. 8d, e; Fig. S8b, c), consistent with previously reported kinetics of monosynaptic transmission from OSNs to STCs[62,69]. EPSC time to peak was also similar between the two genotypes (Fig. 8f; Fig. S8d). These results should be interpreted with caution given the small number of STCs with monosynaptic responses to optogenetic stimulation of Gγ8-ChIEF-Citrine OSNs that were analyzed. Overall, however, the responses evoked by stimulation of Gγ8-ChIEF-Citrine OSNs were kinetically similar to responses evoked by stimulation of OMP-ChIEF-Citrine OSNs, providing clear evidence that immature OSNs form monosynaptic glutamatergic connections with STCs.

## Immature OSNs enable mice to detect and discriminate odors

Having shown that immature OSNs provide odor information to OB neurons, we next asked whether mice can detect and discriminate odors using immature OSNs alone. To answer this question, we needed to generate mice that lacked mature OSNs. Methimazole (MMZ) selectively ablates OSNs but spares OE basal stem cells, enabling OSNs to repopulate the OE over about a month[13,70–72]. We reasoned that at early time points after MMZ administration, the OE would contain

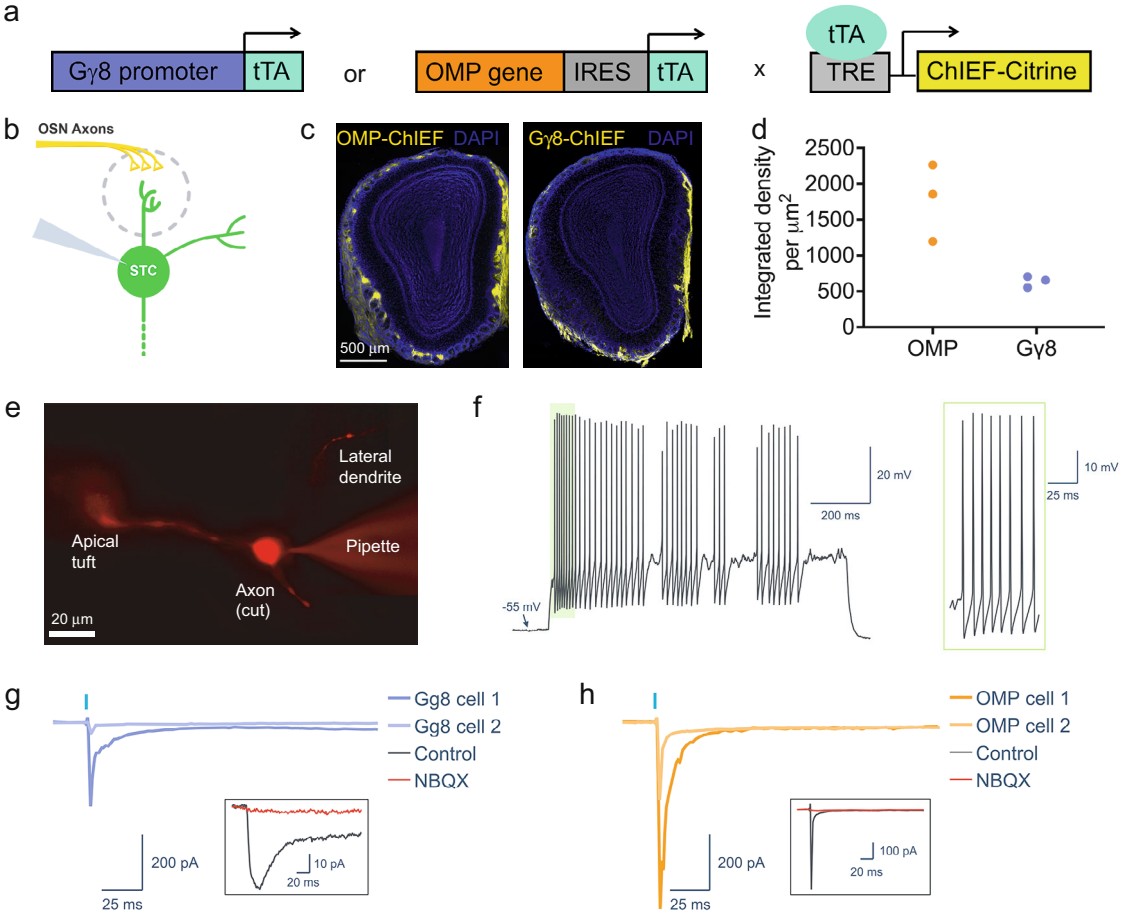

**Fig. 7 | Isolating monosynaptic OSN input to superficial tufted cells. a** Schematic of breeding strategy to generate mice expressing ChIEF-Citrine in either immature or mature OSNs under control of the Gγ8 or OMP promoter, respectively, using the tetracycline transactivator system. **b** Schematic of relevant OB glomerular circuitry. Whole cell voltage-clamp recordings were made from STCs in either Gγ8-ChIEF-Citrine or OMP-ChIEF-Citrine mice. **c** Widefield fluorescence images showing ChIEF-Citrine-expressing axons in an OMP-ChIEF-Citrine and a Gγ8-ChIEF-Citrine mouse. **d** Integrated density of Citrine fluorescence per μm² of the glomerular layer in OMP-ChIEF-Citrine and Gγ8-ChIEF-Citrine mice (n = 3 per group). Symbols: values for individual mice. **e** Widefield fluorescence image of an STC filled with AF594, showing an apical dendrite, a lateral dendrite, and a cut axon. Apical dendrites were

visualized in 13 STCs from OMP-ChIEF-Citrine mice and 6 STCs from Gγ8-ChIEF-Citrine mice. **f** Example of an STC spike train evoked by step current injection, showing an irregular spiking pattern without a depolarizing envelope or rhythmic bursting. **g, h** Recordings from STCs made in normal ACSF containing 20 μM APV (control) and in ACSF containing 10 μM NBQX in addition to APV, demonstrating the presence of monosynaptic input from OSN axons. All traces are averages of 10 trials. **g** Example responses from two STCs elicited by 1 ms 100% intensity light pulse photoactivation of immature Gγ8-ChIEF-Citrine-expressing OSN axons. **h** Example responses from two STCs elicited by 1 ms 100 % intensity light pulse photoactivation of mature OMP-ChIEF-Citrine-expressing OSN axons. Source data are provided as a Source Data file.

immature OSNs but not mature OMP+ OSNs[13]. To validate this model, we performed histological analysis of the dorsal septal OE at 3–7 days after MMZ administration (Fig. 9a). OE width and the number of immature GAP43+ OSNs increased linearly from 3–7 days post-MMZ (Fig. 9b–e). In contrast, OMP+ OSNs were completely absent at 3–6 days post-MMZ (Fig. 9f). At 7 days post-MMZ, two of the three mice still lacked OMP+ OSNs while the third had very sparse OMP + OSNs: 2.6 per mm (Fig. 9g). This equates to only 0.48% of the mean OMP+ OSN density in saline-injected mice, whereas GAP43+ OSN density recovered to 55.9% of the saline-injected value by 7 days post-MMZ. We also collected images of OMP-stained OE turbinates, where we observed the same pattern of no mature OSNs at 3–6 days post-MMZ and sparse OMP-stained OSNs at 7 days post-MMZ (Fig. 9h).

We also wanted to determine whether immature OSN axons reach the OB as early as 5 days post-MMZ. We performed retrograde tracer injections in the glomerular layer of the dorsal anterior OB in mice that had been injected with MMZ 5 days earlier. After perfusion 24 h later, we identified sparse, tracer labeled immature OSNs in the OE (Fig. 9i), which were GAP43+ (Fig. 9j), indicating that at least some 5-day-old OSNs extend their axons into glomeruli. Hence, we concluded that

early time points post-MMZ provide an opportunity to study input provided by immature OSNs in isolation.

We then performed several behavioral assays to determine whether mice can use sensory input from immature OSNs to detect and/or discriminate odors. 8-week-old C57BL/6 J mice received a single intraperitoneal injection of either MMZ or saline and recovered for 3–7 days (Fig. 10a). These mice then performed the buried food assay, which tests the ability of mice to detect volatile odorants[73].

All saline-injected mice successfully located the buried food within 10 min, whereas all mice failed the task at 3 days post-MMZ (Fig. 10b). As early as 5 days post-MMZ, however, 3/12 mice successfully located the buried food (Fig. 10b), and an even larger proportion located the food at 6 days (5/12 mice) and 7 days (7/12 mice) post-MMZ. There was a significant difference between the time to complete the task at 3 vs. 7 days post-MMZ (Fig. 10b). Failure to locate the buried food could not be attributed to any difference in digging behavior between mice that successfully located the buried food vs. those that failed the task (Fig. 10c).

As an independent test of odor detection ability, we used a two-choice odor detection assay to determine whether mice could detect a

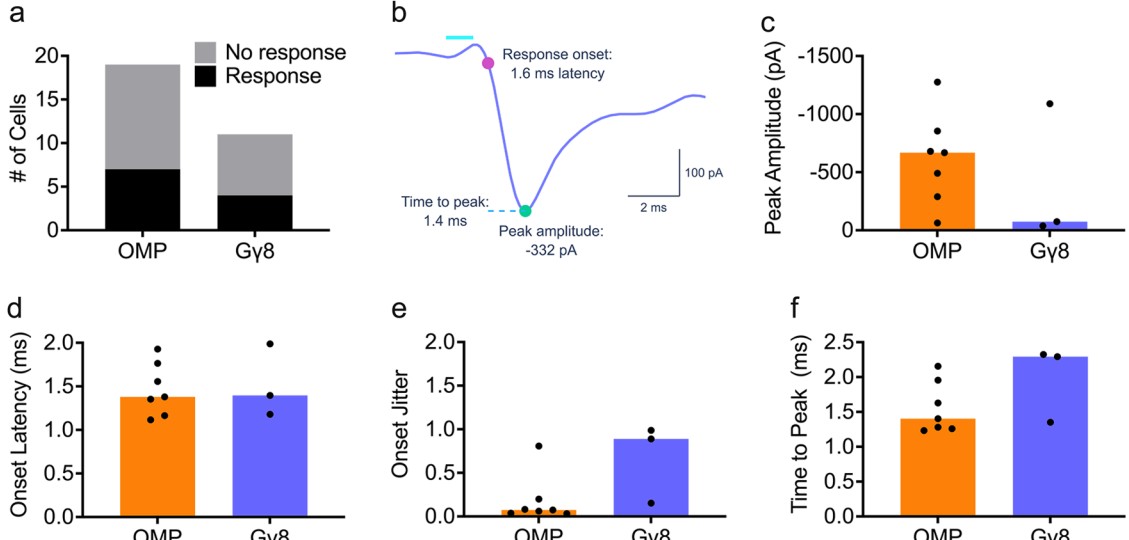

**Fig. 8 | Immature OSNs provide monosynaptic input to superficial tufted cells.**
**a** A similar proportion of recorded STCs from both OMP-ChIEF-Citrine and Gγ8-ChIEF-Citrine mice responded monosynaptically to photoactivation (Fisher's exact test. $P > 0.99$, $n = 19$ cells from 10 OMP-ChIEF-Citrine mice and 11 cells from 5 Gγ8-ChIEF-Citrine mice). One STC from the Gγ8-ChIEF-Citrine group showed a monosynaptic response to optogenetic stimulation with a 1 ms, 50% intensity light pulse, but its dataset was incomplete and lacked sweeps for 1 ms, 100% intensity light stimulation. Data for optogenetic stimulation with a 1 ms, 50% intensity light pulse are shown in Fig. S8. **b** Properties of each light-evoked response analyzed: response onset, time to peak, and peak amplitude. **c–f** Data for optogenetic stimulation with a 1 ms, 100% intensity light pulse. **c** Median peak amplitudes were not significantly different between OMP-ChIEF-Citrine and Gγ8-ChIEF-Citrine mice (Mann–Whitney test. $P = 0.52$, $U = 7$, $n = 7$ cells from 5 OMP-ChIEF-Citrine mice, and $n = 3$ cells from 2 Gγ8-ChIEF-Citrine mice). Bars: median, symbols: values for individual neurons. **d** Median onset latency was not significantly different between OMP-ChIEF-Citrine and Gγ8-ChIEF-Citrine mice (Mann–Whitney test. $P = 0.67$, $U = 8$, $n = 7$ cells from 5 OMP-ChIEF-Citrine mice, and $n = 3$ cells from 2 Gγ8-ChIEF-Citrine mice). Bars: median, symbols: values for individual neurons. **e** Median onset jitter was not significantly different between OMP-ChIEF-Citrine and Gγ8-ChIEF-Citrine mice (Mann–Whitney test. $P = 0.067$, $U = 2$, $n = 7$ cells from 5 OMP-ChIEF-Citrine mice, and $n = 3$ cells from 2 Gγ8-ChIEF-Citrine mice). Bars: median, symbols: values for individual neurons. **f** Median time to peak was not significantly different between OMP-ChIEF-Citrine and Gγ8-ChIEF-Citrine mice (Mann–Whitney test. $P = 0.18$, $U = 4$, $n = 7$ cells from 5 OMP-ChIEF-Citrine mice, and $n = 3$ cells from 2 Gγ8-ChIEF-Citrine mice). Bars: median, symbols: values for individual neurons. All relevant statistical tests were two-tailed. Source data are provided as a Source Data file.

food odorant (peanut butter cookie). Saline-injected mice were all able to detect the food odorant relative to mineral oil (MO), whereas no mice detected the food odorant at 3 days post-MMZ (Fig. 10d, e). At just 5 days post-MMZ, however, 3/6 mice showed significant odor detection, directly paralleling the recovery of olfactory-guided behavior in the buried food test at this time point (Fig. 10b). The proportion of mice showing significant odor detection increased to 4/6 at 6 days and 5/6 at 7 days post-MMZ (Fig. 10d). Furthermore, investigation ratio (time spent sniffing the odorant divided by the total time spent sniffing odorant plus MO) was significantly greater at 5, 6 and 7 days post-MMZ than at 3 days post-MMZ (Fig. 10e). Together, the data from these assays demonstrate that mice can detect odors using immature OSNs alone.

To determine whether sensory input from immature OSNs was sufficient to enable odor discrimination, we performed an odor habituation-dishabituation task[73] (see Methods). As expected, saline-injected mice sniffed less (habituated) to familiar odorants and sniffed more (dishabituated) to novel odorants (Fig. 10f). At 3 days post-MMZ, mice showed little habituation or dishabituation, and both parameters were significantly different to saline-injected mice (Fig. 10f–h). At 5–7 days post-MMZ, both habituation and dishabituation began to recover, with a statistically significant increase in both parameters between 3 vs. 5, 6 or 7 days post-MMZ (Fig. 10g, h). Hence, odor discrimination begins to recover prior to the emergence of mature OSNs.

## Discussion

Our goal in this study was to determine what role, if any, OSNs play in odor processing before they reach maturity. Our data demonstrate that OSNs begin to provide odor input to OB neurons while they are still immature. Crucially, immature and mature OSNs had similar odorant selectivity. However, immature and mature OSNs contributed different information about odorant concentration in vivo. We also employed a pharmacological OSN ablation model to show that mice are able to perform behavioral odor detection and discrimination tasks using sensory input derived only from immature OSNs. Our study suggests that immature OSNs play a previously unappreciated role in olfaction by providing odor input that is functionally distinct from that supplied by mature OSNs. More broadly, our findings provide insight into the mechanisms by which endogenously generated adult-born neurons wire into existing circuits without disrupting their function.

Our slice electrophysiology experiments (Figs. 7, 8; Fig. S8) demonstrate that immature OSNs provide monosynaptic input to STCs. It was surprising that STC response amplitudes were similar in Gγ8-ChIEF-Citrine and OMP-ChIEF-Citrine mice, given that on average, there is a lower density of immature vs. mature OSN axons per glomerulus (Fig. 7c, d) in these mice. However, both the small number of recorded STCs that exhibited monosynaptic responses, and the use of multiglomerular optogenetic stimulation, mean that a more extensive study comprising a larger number of recorded cells, more targeted photoactivation, and perhaps quantification of the density of the stimulated axons, would be required to definitively determine whether there are differences in the amplitude of monosynaptic responses evoked by optogenetic stimulation of immature vs. mature OSN axons. Comparison of the response kinetics following optogenetic stimulation of immature vs. mature OSNs did not reveal significant differences in response latency, jitter or time to peak. Although the number of recorded STCs is small, these comparisons suggest that the monosynaptic input provided by immature OSNs is similar to that from mature OSNs. This finding corroborates and extends our previous in vivo evidence that immature OSN photoactivation evokes robust stimulus-locked firing of neurons in multiple OB layers[4]. Furthermore, our in vivo and ex vivo calcium imaging data demonstrate odor-evoked

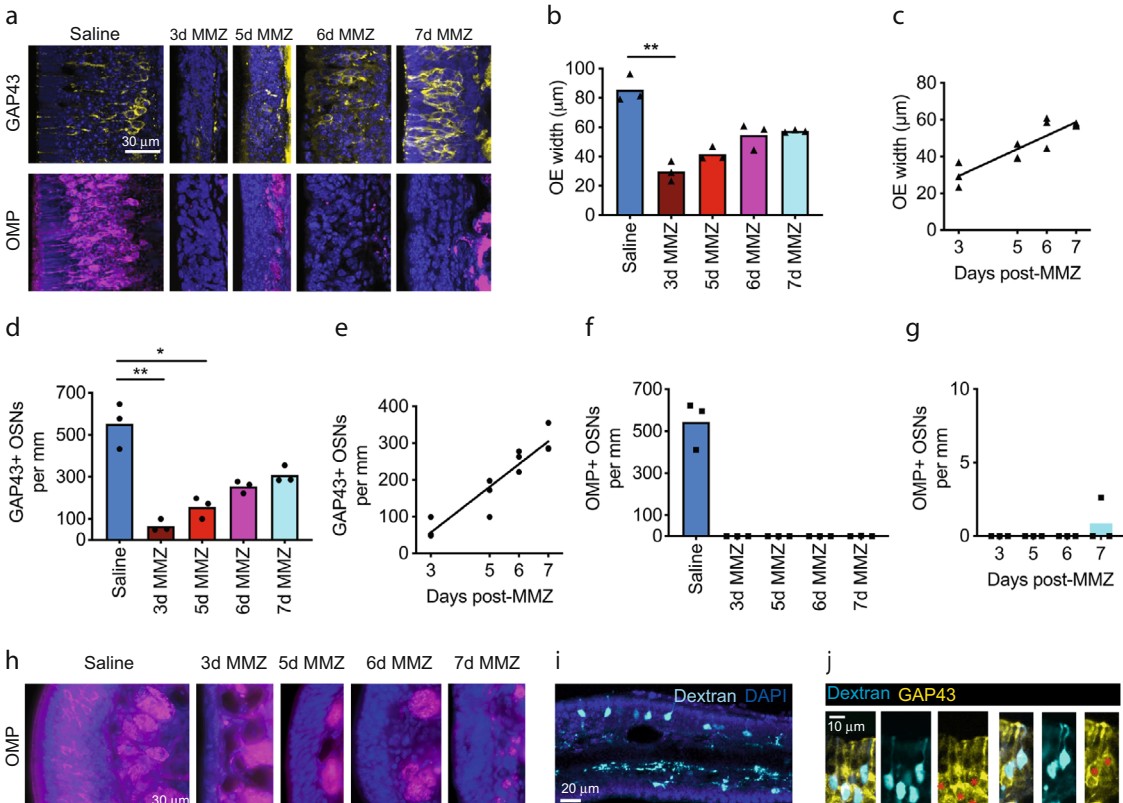

**Fig. 9 | The olfactory epithelium contains immature but not mature OSNs at early time points after methimazole administration. a** MIPs of 2-photon z-stacks of coronal OE sections stained for GAP43 or OMP. **b** OE width is significantly reduced 3 days post-MMZ compared to saline-injected mice (One-way ANOVA on Ranks. P < 0.001, Kruskal−Wallis statistic = 12.4. Dunn's multiple comparisons tests. 3-day, P = 0.004, Z = 3.29; 5d, P = 0.071, Z = 2.37; 6d, P = 0.94, Z = 1.19; 7d, P = 0.68, Z = 1.37; vs. saline. n = 3 mice per group). **c** OE width increases linearly between 3 and 7 days post-MMZ (Linear regression. P < 0.001, R² = 0.80, n = 3 mice per group). **d** GAP43+ OSN linear density is significantly reduced at 3 and 5 days post-MMZ compared to saline-injected mice (One-way ANOVA on Ranks: P < 0.001, Kruskal−Wallis statistic = 13.4. Dunn's multiple comparisons tests. 3d MMZ, P = 0.005, Z = 3.24; 5d MMZ, P = 0.048, Z = 2.51; 6d MMZ, P = 0.40, Z = 1.65; 7d MMZ, P = 1.00, Z = 0.82; vs. saline-injected mice. n = 3 mice per group). **e** GAP43+ OSN

linear density increases linearly between 3 and 7 days post-MMZ (Linear regression. P < 0.001, R² = 0.88, n = 3 mice per group). **f, g** OMP+ OSNs are absent from the septal OE at 3–6 days post-MMZ and very sparse at 7 days post-MMZ. Note different y-axis scales in **f** and **g**. **b–g** Bars: mean per group, symbols: individual mice. **h** Widefield fluorescence images of lateral turbinates from OMP-stained coronal OE sections from mice that received either saline, or MMZ 3–7 d previously. n = 3 mice per group. **i–j** Confocal images from two mice injected with biotin-conjugated dextran at 5 days post-MMZ and perfused 24 h later. **i** MIP of coronal OE section stained with AF555-streptavidin (Dextran, cyan). **j** Single optical sections from different mice showing colocalization of tracer-labeled OSNs (Dextran, cyan) with GAP43 (yellow). Red asterisks: co-labeled cells in GAP43 images. All relevant statistical tests were two-tailed. Source data are provided as a Source Data file.

increases in intracellular calcium in both the somata and axon terminals of immature OSNs (Fig. 2), suggesting that immature OSNs can detect and transduce odorant binding. Taken together, the most likely explanation for our findings is that immature OSNs provide odor input to OB neurons. This capability is consistent with key findings from recent expression analysis studies. First, immature (GAP43- or Gγ8-expressing) OSNs express ORs[35,42,74], with OR expression at both the mRNA and the protein level beginning at 4 days of neuronal age[29], several days prior to the onset of OMP expression[16,27–30]. Second, some immature OSNs express transcripts encoding the signal transduction machinery required for odorant binding to trigger action potential generation[35]. Finally, 5-day-old OSNs generated in young adult mice expressed adenylyl cyclase 3 (AC3) protein[30], which is necessary for olfactory-dependent behavior[75].

However, our data cannot rule out the possibility that ephaptic transmission from mature to immature OSN axons in olfactory nerve fascicles makes some contribution to odor-evoked calcium signals in immature OSN axons in the OB. While there is no direct physiological evidence for ephaptic transmission between OSN axons, a modeling study has suggested that in the olfactory nerve layer, an action potential in one OSN axon could evoke action potentials in other axons within the same fascicle via ephaptic interactions[51]. However, because

the same group found no evidence for gap junctions between OSN axons[76], the structural mechanism by which ephaptic transmission could occur is unclear.

It is also important to consider what an increase in OSN presynaptic terminal calcium concentration signifies. The precise relationship between the number of action potentials and changes in presynaptic terminal calcium concentration is unclear for either immature or mature OSNs and would be very technically challenging to determine directly. However, arrival of an action potential at a mature OSN presynaptic terminal is known to cause Cav2.1 or Cav2.2 channels to open, and the resultant calcium influx triggers synaptic vesicle fusion and glutamate release[77,78]. Immature OSN presynaptic terminals contain synaptic vesicles and are structurally similar to those of mature OSN axons[4]. Furthermore, optogenetic stimulation of immature OSN axons evokes time-locked responses in OB neurons (Figs. 7, 8)[4], suggesting that immature OSN presynaptic terminals are capable of neurotransmitter release.

The relationship between OSN presynaptic calcium concentration and neurotransmitter release is superlinear, meaning that a moderate presynaptic calcium influx strongly modulates the amount of glutamate released[79,80]. Both simultaneous imaging of calcium and neurotransmitter release in mature OSN presynaptic terminals[79], and

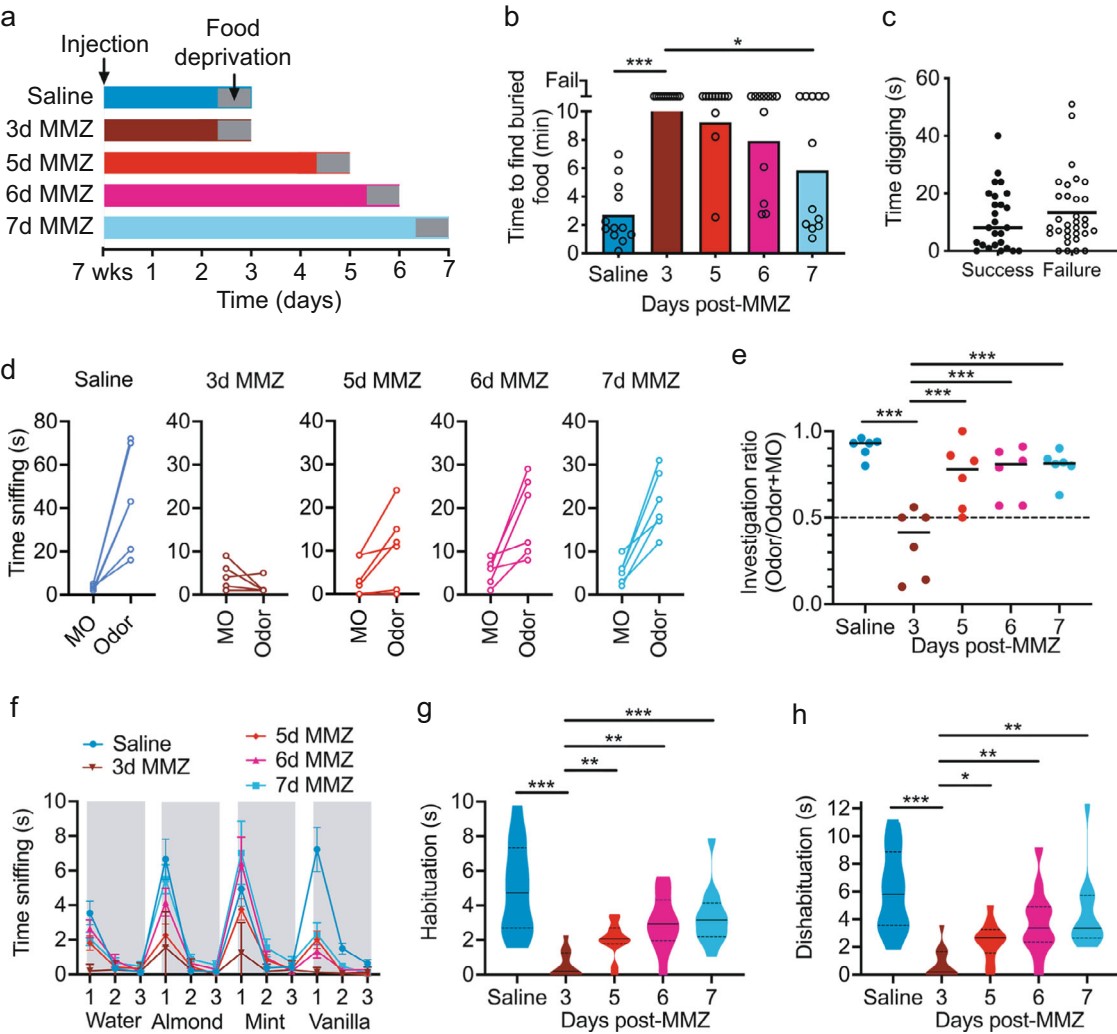

**Fig. 10 | Mice can detect and discriminate odors using only immature OSNs.**
**a** Experimental design. **b** Buried food assay performance improved from 5d-MMZ to 7d-MMZ (One-way ANOVA on Ranks. $P < 0.001$, Kruskal–Wallis statistic = 31.6. Dunn's multiple comparisons. Saline, $P < 0.001$, $Z = 5.10$; 5d-MMZ, $P > 0.99$, $Z = 0.86$; 6d-MMZ, $P = 0.47$, $Z = 1.56$; 7d-MMZ, $P = 0.017$, $Z = 2.86$; all vs. 3d-MMZ, $n = 12$ mice per group). Bars: mean, symbols: individual mice. **c** Similar acclimation period digging time for buried food assay success vs. failure (Mann–Whitney test. $P = 0.44$, $U = 367$, $n = 58$ mice). Lines: median, symbols: individual mice. **d** Time sniffing mineral oil (MO) vs. Nutter Butter (Odor). Symbols: individual mice. **e** Greater investigation ratio in other groups vs. 3d-MMZ (One-way ANOVA. $P < 0.001$, $F_{4,26} = 11.96$. Sidak's multiple comparisons. Vs. saline: 3d-MMZ, $P < 0.001$, $t = 6.43$; 5d-MMZ, $P = 0.20$, $t = 1.88$; 6d-MMZ, $P = 0.20$, $t = 1.73$; 7d-MMZ, $P = 0.23$, $t = 1.24$. Vs. 3d-MMZ: 5d-MMZ, $P < 0.001$, $t = 4.54$; 6d-MMZ, $P < 0.001$, $t = 4.70$; 7d-

MMZ, $P < 0.001$, $t = 5.18$. $n = 6$ mice per group). Lines: mean, symbols: individual mice. **f** Odor habituation-dishabituation assay mean sniffing time ($n = 12$ mice per group). Error bars: standard deviation. **g** Lower habituation time across odorants for 3d-MMZ vs. other groups (Welch's ANOVA. $P < 0.001$, $W_{4,26.4} = 14.68$. Dunnett's multiple comparisons. Saline, $P < 0.001$, $t = 5.58$; 5d-MMZ, $P = 0.003$, $t = 3.94$; 6d-MMZ, $P = 0.001$, $t = 4.63$; 7d-MMZ, $P < 0.001$, $t = 4.99$; vs. 3d-MMZ. $n = 12$ mice per group). **h** Lower dishabituation time across odorants for 3d-MMZ vs. other groups (Welch's ANOVA. $P < 0.001$, $W_{4,26.5} = 11.85$. Dunnett's multiple comparisons. Saline, $P < 0.001$, $t = 5.69$; 5d-MMZ, $P = 0.015$, $t = 3.24$; 6d-MMZ, $P = 0.007$, $t = 3.81$; 7d-MMZ, $P = 0.004$, $t = 4.09$; vs. 3d-MMZ. $n = 12$ mice per group). **g, h** Solid lines: median. Dashed lines: quartiles. Statistical tests were two-tailed. Source data are provided as a Source Data file.

electrophysiological recordings of rodent external tufted cell and juxtaglomerular neuron responses to olfactory nerve stimulation in varying extracellular calcium concentrations[79,80], demonstrated very similar Hill coefficients of close to 2. Because olfactory nerve stimulation would be expected to have triggered action potentials in both immature and mature OSN axons, the strong agreement in the Hill coefficient derived via electrophysiology and imaging methods suggests that there may also be a superlinear relationship between presynaptic calcium influx and glutamate release in immature OSN presynaptic terminals. However, this remains to be directly determined.

We also found that immature OSN axons did not respond to a larger number of odorants compared to their mature counterparts (Fig. 4). This is surprising given that single cell RNA sequencing studies

have indicated that a subset of immature OSNs, including some late immature OSNs that possess transcripts for odorant signal transduction machinery, transcribe multiple OR genes[35,42]. If these immature OSNs also express multiple ORs at the protein level, then they would be expected to respond to a larger number of odorants[37,38]. We believe that our experimental design would have been able to detect reduced odorant selectivity in Gγ8-GCaMP6s mice, had it been present: almost 50% of imaged glomeruli responded to more than one of the odorants in our panel, indicating good coverage with potential activating odorants for the dorsal surface region that we imaged. Furthermore, the low baseline standard deviation in our images enabled us to detect even weak odorant responses with $\Delta F/F$ values as low as 1.5% for individual glomeruli. Interestingly, immature OSN axons in lateral M72 glomeruli responded to a maximum of two of the eight tested

odorants, whereas mature OSN axons in lateral M72 glomeruli responded to up to four of the tested odorants. This discrepancy might be due to differences in the threshold for detection of these additional odorants by immature vs. mature OSNs, or to the sparser innervation of glomeruli by immature vs. mature OSNs (Fig. 2a). Additional experiments involving imaging of a larger number of M72 glomeruli and using a range of odorant concentrations would be required to test these possibilities. Overall, however, our data from Gγ8-GCaMP6s-M72-RFP and OMP-GCaMP6s-M72-RFP mice support our conclusion that the odorant selectivity of immature OSNs is not lower than that of mature OSNs at the level of the glomerulus.

What underlies odorant selective glomerular responses in Gγ8-GCaMP6s mice? First, there may be a regulatory mechanism that prevents translation or dendritic and axonal trafficking of more than one OR. Second, low-level expression of additional ORs, as is the case for many of the multi-OR-expressing immature OSNs identified in RNA sequencing studies[35,42], may have little impact on glomerular odorant responses. Given the massive convergence of OSN axons expressing each OR to form a small number of glomeruli[60,81,82], and the large number of possible combinations of ORs that could be co-expressed (even accounting for regional bias within the OE in terms of which OR(s) are selected[35]), an additional OR expressed at a low level by any individual immature OSN is likely to only make a tiny, perhaps even undetectable, contribution to immature OSN input to a glomerulus. Third, OSNs expressing additional ORs at levels sufficient to perturb axon guidance[38,40,41], such as those expressing two ORs at similar levels[29,35,42], may not survive. Given both the large number of possible OR combinations and the well-established finding that there are only ~2 glomeruli per OR in each OB[41,82–84], the axons of these multi-OR-expressing OSNs would be expected to project to ectopic locations where they are unlikely to form synaptic connections, and hence, survive[85,86].

How does the previously unidentified odor input stream provided by immature OSNs affect olfactory function? While immature OSNs are odorant selective, they provide different concentration information compared to mature OSNs, at least for the odorants and concentration range that we tested. The percentage of glomeruli that responded to each concentration of each odorant was similar in Gγ8-GCaMP6s and OMP-GCaMP6s mice (Fig. 3c). However, the amplitude and latency of immature OSN responses were dependent on concentration across the entire range that we tested, whereas mature OSN responses were only significantly different between lower concentrations. This suggests that immature OSNs provide unique concentration information to glomeruli, which may enable mice to discriminate between higher concentrations of odorants. These differences in the concentration dependence of odorant responses in immature OSNs could be due to lower immature OSN intrinsic excitability, lower efficiency transduction of odorant binding, altered action potential propagation, and/or greater GABA$_B$ receptor-mediated presynaptic inhibition[87] at immature OSN presynaptic terminals. Any of these mechanisms, or a combination, could result in a more graded immature OSN response to increasing odorant concentration, whereas mature OSNs are already maximally recruited at concentrations midway through the range that we tested.

In order to test whether odor input provided by immature OSNs is behaviourally relevant, we established a MMZ ablation-based model wherein mice at early time points post-MMZ possess immature but not mature OSNs (Fig. 9). We did not detect any mature OSNs in the OE at 3–6 days post-MMZ, suggesting that OSN ablation was highly effective. Although we quantified OSN density in the septal OE, we observed the same pattern in the turbinates. Similarly, previous studies in mice have shown virtually complete ablation of either OMP-stained OSNs or OSNs expressing a GFP-tagged OR[12,13,88]; the only study to quantify this throughout the OE found 99.8% ablation of MOR28-expressing OSNs[88]. In contrast, an earlier study in rats found some intact ventral and

lateral regions of OE after lower doses of MMZ, with complete destruction of the olfactory mucosa requiring a higher dose than was used in this study[89]. Several factors could contribute to this discrepancy. First, OSNs may have been lost from regions of the OE that appeared intact in low magnification images of hematoxylin and eosin-stained sections in Genter et al.[89]. Second, histological analysis was performed at 24 h post-MMZ in Genter et al.[89], rather than at 3–5 days post-MMZ as in this and other mouse studies[12,13,88], by which time further damage may have been evident. Finally, there may be differences in the completeness of OSN ablation between mice and rats.

Importantly, we found that mice were unable to detect or discriminate odors 3 days after MMZ administration, ruling out the possibility that any very sparse residual mature OSNs that had survived MMZ treatment could mediate these behaviors. Hence, our data provide strong evidence that odor input from newly generated immature OSNs is sufficient to mediate odor detection (Fig. 10b, e). Just 5–6 days after MMZ administration, a subset of mice successfully performed the buried food and two-choice odor detection assays, consistent with AC3 expression in 5-day-old OSNs[30]. We also found that the ability to discriminate odors began to recover at 5 days post-MMZ. Additionally, we found using retrograde tracer injections that the axons of some immature OSNs innervated the glomerular layer as early as 5 days post-MMZ (Fig. 9i, j), providing a substrate for odor input to the OB that can mediate odor-guided behavior. Studies using Ascl-1-driven labelling of basal stem cells[29,30] indicated that robust innervation of glomeruli by OSN axons does not occur until 8–10 days after terminal cell division. Nonetheless, images from both of these studies suggest that very sparse glomerular layer innervation by OSNs as young as 5 days does occur, in agreement with our data.

At 5–6 days post-MMZ, no mature OSNs were detectable in the septal OE by OMP immunohistochemistry (Fig. 9f,g). We did detect a very low density of mature OSNs (2.6 per mm) in just one of the three mice analyzed at 7 days post-MMZ. However, this very restricted population of mature OSNs is unlikely to account for the behavioral recovery seen at 7 days post-MMZ for several reasons. First, the ability to detect and discriminate odors was already present at 5 days post-MMZ in a subset of mice and did not appear suddenly at 7 days post-MMZ. Second, mature OSNs were absent from the septal OE of the other two 7-day post-MMZ mice analyzed. Finally, this very small number of mature OSNs represents just 0.48% of the mean density of mature OSNs in our saline-injected control mice. In contrast, immature OSN density increased linearly across 3–7 days post-MMZ, reaching 56% of the mean saline-injected control value by 7 days post-MMZ. Hence, our data suggest that odor input from immature OSNs supports both odor detection and at least relatively simple odor discrimination. Notably, performance on the odor detection and discrimination assays did not fully recover by 7 days post-MMZ relative to the saline control group. This is not surprising given that saline control mice receive odor input not only from a larger number of immature OSNs but also from a high density of mature OSNs.

The time course of appearance of OMP+ OSNs following MMZ treatment in this study closely matches that described previously for both the healthy mouse OE[16,27–30] and the mouse OE following MMZ-mediated ablation[13]. Hence, the time course of OSN maturation following MMZ treatment is similar to that seen during constitutive OSN neurogenesis. However, in the healthy adult olfactory system, a large number of mature OSNs project to each glomerulus, meaning that while immature OSNs can provide odor input sufficient to mediate odor detection and simple odor discrimination, they need not perform this task alone. What role, then, do they play?

We propose that immature OSNs provide a functionally distinct information stream to OB neurons that enables discrimination of higher odorant concentrations than does mature OSN input. In the healthy olfactory system, therefore, immature and mature OSNs provide complementary odor information to individual glomeruli, which

may be beneficial by simultaneously enabling discrimination across a wide concentration range and reliable responses to low odorant concentrations, respectively. We have also shown previously that both formation and elimination of presynaptic terminals in glomeruli occurs at a much higher rate for immature OSN axons than for mature OSN axons[4]. Moreover, this presynaptic terminal turnover was dramatically reduced when sensory input was blocked via naris occlusion[4]. This accelerated activity-dependent structural plasticity enables immature OSNs to adapt rapidly to novel sensory experience and may play an important role in the integration of newly generated OSNs into OB circuits, and hence their survival.

Sensory information provided by immature OSNs is therefore likely to make the most important contribution under circumstances when few mature OSNs are present; namely, during initial wiring of glomerular circuits or during regeneration after a significant proportion of pre-existing OSNs have been lost, such as following a viral insult or the experimental administration of MMZ. The ability of immature OSNs to encode information about high concentration odorants may be important in these scenarios to enable odor-guided behaviors essential for organismal survival.

## Methods

### Experimental animals

All animal procedures conformed to National Institutes of Health guidelines and were approved by the Carnegie Mellon University and University of Pittsburgh Institutional Animal Care and Use Committees. C57BL/6J mice (strain #000664) were purchased from the Jackson Laboratory and all other lines were bred in-house. Mice were maintained on a 12 h light/dark cycle in individually ventilated cages at 22 °C and 48% humidity with unrestricted access to food and water unless otherwise stated. Mice were group-housed if same sex littermates were available. A total of 182 mice were used in this study.

Generation of the Gγ8-tTA[90], OMP-IRES-tTA[91], tetO-ChIEF-Citrine[4], tetO-GCaMP6s[92], tetO-sypGFP-tdT[93], OMP-Cre[94], Ai9 [flox-tdTomato][95] and M72-IRES-RFP[58] mouse lines has been published. Mice for 2-photon calcium imaging experiments were Gγ8-GCaMP6s [Gγ8-tTA$^{+/-}$;tetO-GCaMP6s$^{+/-}$], OMP-GCaMP6s [OMP-tTA$^{+/-}$;tetO-GCaMP6s$^{+/-}$], Gγ8-GCaMP6s-OMP-tdT [Gγ8-tTA$^{+/-}$;tetO-GCaMP6s$^{+/-}$;OMP-cre$^{+/-}$;flox-tdT$^{+/-}$], OMP-GCaMP6s-OMP-tdT [OMP-tTA$^{+/-}$;tetO-GCaMP6s$^{+/-}$;OMP-cre$^{+/-}$;flox-tdT$^{+/-}$], Gγ8-GCaMP6s-M72-RFP [Gγ8-tTA$^{+/-}$;tetO-GCaMP6s$^{+/-}$;M72-IRES-RFP$^{+/+}$] or OMP-GCaMP6s-M72-RFP [OMP-tTA$^{+/-}$;tetO-GCaMP6s$^{+/-}$;M72-IRES-RFP$^{+/+}$]. Note that homozygosity of the M72-RFP allele was necessary for in vivo visualization of M72 glomeruli. Images from Gγ8-sypGFP-tdT [Gγ8-tTA$^{+/-}$;tetO-sypGFP-tdT$^{+/-}$] and OMP-sypGFP-tdT [OMP-tTA$^{+/-}$;tetO-sypGFP-tdT$^{+/-}$] mice were collected during a previous study[4]. Mice for electrophysiology experiments were Gγ8-ChIEF-Citrine [Gγ8-tTA$^{+/-}$;tetO-ChIEF-Citrine$^{+/-}$] or OMP-ChIEF-Citrine [OMP-tTA$^{+/-}$;tetO-ChIEF-Citrine$^{+/-}$]. Mice for behavioral experiments were C57BL/6J (The Jackson Laboratory). All genetically modified mice were of mixed 129x C57BL/6J background, and each experimental group comprised approximately equal numbers of male and female mice, which were randomly assigned to experimental groups. Mice were genotyped by PCR using previously validated primers (Supplementary Dataset 1)[4,92,94,95].

### In vivo 2-photon calcium imaging and data analysis

P21-23 mice were anesthetized with ketamine/xylazine (100/10 mg/kg), ketoprofen (5 mg/kg) was administered and ~1 mm diameter craniotomies were made over one or both OBs and cranial windows were implanted[4]. A head bar was also attached. Mice were imaged with either a VIVO 2-photon microscope (3i) using a Plan-Apo 20x/1.0NA water-immersion objective (Zeiss) and a Chameleon Ultra II IR laser (Coherent) mode-locked at 935 nm using SlideBook (3i); or a Bergamo II 2-photon microscope (ThorLabs) using a SemiApo 20x/1.0NA water-immersion objective (Olympus) and an Insight X3 IR laser (Newport)

mode-locked at 935 nm using ThorImage 4.0 (ThorLabs). Time-lapse images of a single optical section were collected at 10.17 frames/s on the VIVO system (266 μm², pixel size 1.04 μm) or 15 frames/s on the Bergamo system (328 μm², pixel size 0.64 μm). Top-up doses of ketamine/xylazine (50/5 mg/kg) were given during the imaging session as required to maintain light anesthesia.

Odorant stimulation was performed using a custom-built Arduino-controlled olfactometer. Stimulation and image acquisition timing were controlled and recorded using either Igor Pro (Wavemetrics) or Python, ThorSync and ThorImage (ThorLabs). The seven-odorant panel consisted of odorants known to activate the dorsal surface of the OB: ethyl butyrate, hexanal, 2-hexanone, propionic acid, isoamyl acetate, methyl salicylate and acetophenone (all 1% v/v dilutions in mineral oil). The concentration panel consisted of 0.5, 1, 5 and 10% v/v dilutions of ethyl butyrate, isoamyl acetate, hexanal, propionic acid and 2-hexanone. Preliminary experiments (Fig. 1) used 1% ethyl butyrate, hexanal, isoamyl acetate and 2-hexanone. Saturated vapor from odorant vials, or a blank stimulus consisting of deodorized dehumidified air, were delivered into an oxygen carrier stream at a constant flow rate of 1 l/min. The air puff stimulus used to test for putative mechanosensory responses consisted of a 1 s deodorized dehumidified air puff delivered at a flow rate of 1 l/min in the absence of a carrier stream. All stimuli were 1 s in duration and were delivered via a tube positioned 5 mm from the opening of the external nares. All image trials were 20 s in duration (4 s baseline, 1 s stimulus, 15 s post-stimulus). Three trials were collected for each stimulus, with odorants and blank stimuli delivered in a pseudo-random order with 90 s between trials.

Data were analyzed using Fiji (ImageJ)[96]. For each imaged region of interest, glomeruli were manually outlined, and the same outlines were used for analysis of all stimulus presentations. Mean fluorescence intensity over time was extracted for each glomerulus for each trial. Data from the three trials were then averaged for each stimulus. The fluorescence intensity change ($\Delta F$) was calculated as the difference between a 1 s baseline prior to stimulus onset and the peak fluorescence within 3 s after stimulus onset. A glomerulus was defined as responding to a stimulus if the $\Delta F$ amplitude was greater than three standard deviations of the baseline period. $\Delta F/F$ (%) was reported as (baseline − peak)/baseline x 100 for significant responses only. For blank subtraction, the $\Delta F/F$ in response to the blank (deodorized air) stimulus for a glomerulus was subtracted from the $\Delta F/F$ in response to each odorant.

Saturation of the GCaMP6s sensor could confound our analysis of response amplitudes. If saturation did occur, responses would be expected to show a rapid onset plateau in fluorescence traces. Each odorant response for each glomerulus was visually inspected, and no evidence of plateaus was observed. We also performed additional analysis of the largest amplitude responses (those with $\Delta F/F$ values >1000%, $n = 17$ glomeruli), in which saturation was most likely to occur. For each of these glomeruli, the maximal fluorescence response was compared to a smaller amplitude response evoked by a lower concentration of the same odorant. For each of these responses ($n = 34$), the change in fluorescence intensity ($\Delta F = F_t - F_{t-1}$) was calculated for each pair of frames. The number of frames between the maximal positive change in $\Delta F$ value (i.e., the maximum rising rate) and the first zero or negative $\Delta F$ value (which would correspond to the onset of response decay or of a plateau) was determined. We reasoned that if GCaMP6s saturation was responsible for the peak fluorescence intensity, responses would transition from a rapid rising phase to a negative or zero $\Delta F$ value within 1–2 frames. In contrast, responses that did not exhibit saturation would gradually transition from positive to negative $\Delta F$ values.

For analysis of mean odorant response amplitude per mouse, all glomeruli that responded to at least one concentration of a particular odorant were included. Because not all included glomeruli responded

to all concentrations of an odorant, values of zero were present for some concentrations. For analysis of the latency and time to peak of odorant responses, we also compared mean values per mouse. For each glomerulus, differences in latency and time to peak were calculated for ascending pairs of odorant concentrations (0.5–1%, 1–5% and 5–10%) only if that glomerulus responded to both concentrations in a pair. For each ascending pair of odorant concentrations, the mean of all glomeruli for which this value was available was then calculated for each mouse.

Lifetime sparseness ($S_L$) provides a measure of the degree to which a glomerular response was attributable entirely to one odorant (highly selective, $S_L = 1$) versus equally distributed across all odorants ($S_L = 0$). $S_L$ was calculated as $(1 - (\Sigma_{i=1,n}(r_i/n)^2)/(\Sigma_{i=1,n}(r_i^2/n)))/(1 - 1/n)$, where $r_i$ is the $\Delta F/F$ of the glomerular response to odorant $i$ and $n$ is the total number of odorants[56].

### Glomerular density analysis
For each Gγ8-sypGFP-tdT and OMP-sypGFP-tdT mouse, analysis was performed on a single optical section from the center of a 2-photon z-stack through the depth of the glomerular layer[4]. For each Gγ8-GCaMP6s and OMP-GCaMP6s mouse, maximum intensity projections (MIPs) of single optical section 2-photon time series showing responses to stimulation with the 7-odorant panel were analyzed for one field of view. Glomeruli were identified and density per mm$^2$ was calculated.

### Ex vivo 2-photon calcium imaging and data analysis
P21-23 mice Gγ8-GCaMP6s mice were deeply anesthetized with isoflurane and decapitated into ice-cold oxygenated modified Ringer's solution (in mM: 113 NaCl, 25 NaHCO$_3$, 20 HEPES, 20 glucose, 5 KCl, 3 MgCl$_2$, and 2 CaCl$_2$, pH 7.4). The head was hemisected sagitally and the septum was removed to expose the turbinates. The hemi-head preparation recovered in cold modified Ringer's solution for 40 min prior to imaging.

For 2-photon imaging, one hemi-head per mouse was mounted in a glass dish and was continuously perfused with Ringer's solution at room temperature at 0.75 l/min. A large (673 × 673 µm) field of view containing multiple turbinates was imaged. 1 M stock solutions of odorants (ethyl butyrate, isoamyl acetate, hexanal and 2-hexanone) were made in dimethyl sulfoxide (DMSO), then diluted to 100 µM in Ringer's solution (final DMSO concentration of 0.01%). Odorant solutions were applied for 30 s with 2.5 min intervals between stimuli[97]. Images were collected at 15 frames/s on the Bergamo 2-photon system using 935 nm excitation. Ringer's solution containing 80 mM KCl was applied at the end of each experiment to confirm the viability of the preparation.

For analysis, 50 immature OSNs per mouse were randomly selected and outlined for each odorant, and mean fluorescence intensity over time was extracted for each OSN. $\Delta F/F$ was calculated as for in vivo GCaMP6s data. OSNs were defined as showing a significant response to an odorant if the $\Delta F$ amplitude was greater than three standard deviations of the baseline period.

### Slice electrophysiology and optogenetic stimulation
P18-25 Gγ8-ChIEF-Citrine and OMP-ChIEF-Citrine mice were deeply anesthetized with isoflurane and decapitated into ice-cold oxygenated artificial cerebrospinal fluid (ACSF). The olfactory bulbs were dissected, and sagittal slices (310 µm thick) were prepared using a vibratome (Ci 5000 mz2; Campden Instruments). Slices recovered in ACSF at 35 °C for 20 min. ACSF contained (in mM): 125 NaCl, 25 glucose, 2.5 KCl, 25 NaHCO$_3$, 1.25 NaH$_2$PO$_4$, 1 MgCl$_2$, and 2.5 CaCl$_2$, pH 7.4. Slices were then incubated in ACSF at room temperature until recording. Slices were continuously superfused in ACSF at 35 °C while recording. Current clamp and voltage clamp recordings were made using electrodes filled with (in mM): 120 K-gluconate, 2 KCl, 10 HEPES, 10 Na-phosphocreatine, 4 Mg-ATP, 0.3 Na$_3$GTP, 0.2 EGTA and 0.025

Alexa Fluor 594 (AF594). Whole-cell patch-clamp recordings were made using a Multiclamp 700A amplifier (Molecular Devices) and an ITC-18 acquisition board (Instrutech) controlled by Igor Pro (WaveMetrics). Superficial tufted cells (STCs) were identified under an upright microscope (SliceScope, Scientifica) with IR-DIC by their shape and location in olfactory bulb laminae. Their identity was confirmed by visualization of AF594-filled lateral dendrites and/or spike patterns characteristic of STCs (Fig. 7e, f)[62,63]. STCs close to glomeruli innervated by ChIEF-Citrine expressing axons were selected for recording. We recorded approximately equal numbers of STCs from the dorsal and the ventral surfaces of the OB for both genotypes (Gγ8-ChIEF-Citrine: 6 dorsal and 5 ventral, OMP-ChIEF-Citrine: 10 dorsal, 9 ventral). This enabled us to corroborate our in vivo calcium imaging data, which were obtained from the dorsal OB, and to obtain recordings from the ventral OB, which receives robust innervation by immature OSN axons at this age[4].

Pharmacological agents used were the NMDA receptor antagonist DL-2-amino-5-phosphonovaleric acid (APV, 20 µM), and the AMPA receptor antagonist 2,3-dihydroxy-6-nitro-7-sulfamoyl-benzo(F)quinoxaline (NBQX, 10 µM). Cells were excluded from analysis if their resting membrane potential was depolarized above −45 mV. At least 10 min elapsed between cell selection and the start of photoactivation experiments. For photoactivation of immature or mature OSN axons, slices were illuminated using a 490 nm LED (pe-100; CoolLED Ltd) passed through a 40x/0.8NA water-immersion objective (Olympus) centered on the glomerular layer. An open field stop was used to enable multiglomerular activation.

Light evoked EPSCs were isolated in the presence of APV. We generated a power curve for each STC by systematically increasing the LED intensity (from 10 to 20, 50, 80 and 100 %) for each of three light stimulus durations (0.25, 1 and 2 ms) (Fig. S7). 10 trials per stimulus condition were recorded for each cell, although we could not collect a complete data set for every recorded STC. NBQX was applied at the end of a subset of recordings to confirm that the recorded responses were AMPA-mediated.

### Electrophysiology data analysis
For analysis of our recordings, we selected the shortest stimulus (1 ms at 50% or 100% intensity) that reliably and consistently evoked responses and used data from these trials for all subsequent analysis. Data for 1 ms, 100% intensity stimulation are shown in Fig. 8c–f, and data for 1 ms, 50% intensity stimulation are shown in Fig. S8. For each recording, the mean baseline over a 350 ms window before stimulus onset was subtracted from the recorded current trace. EPSC peak amplitude was defined as the most negative value of the baseline-subtracted current trace in a 250 ms window after stimulus onset for each trial. The response onset latency was defined as the time interval from optogenetic stimulus onset to the time at which the current trace reaches 5% of its peak value. Jitter was calculated by taking the standard deviation of the onset latencies of all trials collected using the same stimulus parameters[98]. The time to peak was defined as the interval of time from response onset to the time of the EPSC peak. A cell was classified as having a response if the EPSC peak amplitude exceeded three standard deviations of the pre-stimulus baseline current. Only STCs showing responses with an onset latency shorter than 2 ms were defined as receiving monosynaptic input from OSN axons[62,69]. A total of 7 STCs from 5 OMP-ChIEF-Citrine mice, and 4 STCs from 3 Gγ8-ChIEF-Citrine mice showed monosynaptic responses to optogenetic stimulation. In the dataset from Gγ8-ChIEF-Citrine mice, one STC showed consistent light-evoked monosynaptic responses, but its power curve was incomplete and lacked response sweeps for the 1 ms duration and 100% intensity stimulus parameter. Therefore, we included this cell in the response counts and the analysis in Fig. S8, but not in the amplitude and kinetic analysis in Fig. 8c–f due to stimulus parameter mismatch. In the dataset from OMP-ChIEF-Citrine mice, one

STC showed consistent light-evoked monosynaptic responses, but its power curve was incomplete and lacked response sweeps for the 1 ms duration and 50% intensity stimulus parameter. This STC was included in the response counts and in Fig. 8c–f but not in Fig. S8.

## Methimazole ablation of OSNs

8-week-old Gγ8-ChIEF-Citrine, OMP-tdT and C57BL6/J mice received a single intraperitoneal injection of either methimazole (MMZ; 75 mg/kg in sterile saline) or an equivalent volume of saline. Behavioral assays or perfusions were conducted 3–7 d later. Time points were accurate to ±1 h at the start of the behavioral assay(s) or at the time of perfusion.

## Retrograde tracer injections

Two P56-P60 C57BL/6 J mice were injected with MMZ. 5 d later, mice were anesthetized with isoflurane (2% in 1 l/min $O_2$), ketoprofen (5 mg/kg) and dexamethasone (2 mg/kg) were administered and ~0.5 mm diameter craniotomies were made bilaterally over the anterior dorsal OB. Approximately 1 µl of a 12 mM solution of dextran (3 kDa, tetra-methylrhodamine and biotin labeled, D7162, Thermo Fisher Scientific) in sterile DMSO was injected using a pulled glass pipette (tip diameter ~30 µm) using a picospritzer (~5 psi, 50-100 ms pulses; General Valve). The pipette was left in place for an additional 5 min, then withdrawn slowly. A thin piece of silicone-based organic polymer (polydimethylsiloxane) was glued over the craniotomy and the scalp was sutured. Mice were transcardially perfused 24 h later.

## Odor detection and discrimination assays

A buried food assay[73] and two-choice odor detection assay[99] were used to test odor detection. An odor habituation-dishabituation assay[73] was used to test odor discrimination. Mice were transported to the behavioral testing room at least 30 min prior to commencing the assay(s). All trials were videoed.

8-week-old C57BL/6J mice performed the buried food and habituation-dishabituation assays (12 mice per group, injected with MMZ or saline 3–7 d prior to testing). Protocols were based on a previously published method[73], with some refinements. Mice were food deprived for 16 h prior to the assays. Videos were collected from two angles for analysis. Each mouse that performed the buried food assay received a single Froot Loop (Kellogg's) for odorant familiarization at the start of food deprivation. All mice consumed this Froot Loop. For the buried food assay each mouse had 5 min to acclimate to the test cage, which contained a 1.5 cm depth of Sani-Chips bedding (P.J. Murphy). Time digging during this 5 min session was quantified from videos. The mouse was then briefly removed, a Froot Loop was buried near the center of the cage, and the mouse was returned to the test cage. The time to locate the buried Froot Loop was then measured; mice failed the task if they had not located the buried food within 10 min.

For the odor habituation-dishabituation assay each mouse acclimated for 30 min in the test cage, which contained a dry filter paper square in a shallow plastic dish that the mice could not directly contact. Mice were then presented with a series of odorant-soaked filter paper squares for 2 min each. The time spent sniffing each odorant (scored as nose oriented toward, and within 1 inch of, the filter paper) was recorded from videos. The odorant series consisted of three trials each of water, almond, mint and vanilla, presented in a fixed order. These are neutral odorants that elicit only modest levels of sniffing[100,101]. Odorants were food extracts diluted 1:100 in water. Habituation was quantified as the difference in sniffing time between the first and second trial with each odorant. Dishabituation was quantified as the difference in sniffing time between the final trial of an odorant and the first trial of the subsequent odorant.

A separate group of 8-week-old C57BL/6 J mice performed the two-choice odor detection assay (6 mice per group). Male and female mice were housed in groups of three. For two consecutive days, each mouse was transferred to a clean cage and given 1.5 g Nutter Butter

cookie (Nabisco). Mice were returned to their home cage once they had eaten the cookie. The next day, mice received a saline or MMZ injection. Mice were food deprived for 16 h prior to testing, which occurred 3–7 d after saline or MMZ injection. Each mouse was transferred to a test cage and given 10 min to acclimate. Mice then had 10 min to investigate filter paper squares odorized with mineral oil (MO) or Nutter Butter cookie suspended in mineral oil (Odor). Mice could not make direct contact with the filter paper squares. The total time spent sniffing each filter paper square was scored manually from videos. Investigation ratio was calculated as the time spent sniffing Odor divided by the total time spent sniffing Odor plus MO, i.e., a value of 0.5 indicates no odor detection. An investigation ratio of 0.75 (i.e., duration sniffing Odor was three times that sniffing MO) was defined as indicating significant odor detection.

## Perfusion, immunohistochemistry and image analysis

Mice were anesthetized with either 200 mg/kg ketamine and 20 mg/kg xylazine or 5% isoflurane in 1 l/min $O_2$ and transcardially perfused with ice-cold phosphate-buffered saline (PBS) followed by 4% paraformaldehyde (PFA) in PBS. OBs and/or OEs were dissected and postfixed overnight before cryopreservation overnight in 30% sucrose, embedding in 10% gelatin, and overnight fixation/cryopreservation in 15% sucrose/ 2% PFA. 40 µm OB sections and 50 µm OE sections were cut coronally using a cryostat. All sections were mounted with Vectashield containing DAPI (Vector Labs). Detailed antibody information is provided in Supplementary Dataset 1.

To quantify co-expression of OMP in P21-23 Gγ8-GCaMP6s-expressing OSNs, OE sections (3 per mouse at 25%, 50% and 75% along the anterior-posterior axis) were stained to detect OMP (anti-OMP primary antibody [1:5000, 96 h at 4 °C, catalog #544-10001, Wako Chemicals] and donkey anti-goat-Alexa Fluor 546 secondary antibody [1:500, 1 h at room temperature]), and GCaMP6s (GFP-Booster-Atto-488 [1:500, 1 h at room temperature, gba-488-100, Chromotek]). Confocal z-stacks of ~1 mm of dorsal septal OE were collected (voxel size, $0.13 \times 0.13 \times 0.5$ µm) using an A1R confocal microscope equipped with an Apo 60x/1.4NA oil immersion objective and Elements software (Nikon). Numbers of Gγ8-GCaMP6s+ and Gγ8-GCaMP6s+OMP+ OSNs were quantified and the percentage of Gγ8-GCaMP6s+OMP+ OSNs determined in each section for three mice. Gγ8-GCaMP6s+ OMP+ OSNs were defined as those that expressed GCaMP6s throughout the soma and OMP throughout the cytosol. To confirm that we did not miss Gγ8-GCaMP6s+OMP+ OSNs for which fluorescence was dim in one or both channels, we quantified the cytosolic fluorescence intensity in both channels for a subset of OSNs that we had classified as Gγ8-GCaMP6s+OMP+, Gγ8-GCaMP6s+OMP– and Gγ8-GCaMP6s-OMP+ (Fig. S1).

Citrine fluorescence was quantified for 3 OB sections per mouse (left and right OBs at 25%, 50% and 75% along the anterior-posterior axis) for Gγ8-ChIEF-Citrine and OMP-ChIEF-Citrine mice ($n = 3$ per group). Images (pixel size 0.64 µm) of entire sections were collected using an Eclipse 90i large area-scanning widefield microscope equipped with a Plan-Apo 10x/ 0.45NA air objective and Elements software (Nikon). Camera settings were the same for all images. Total integrated fluorescence intensity in the Citrine channel was normalized to the area of the glomerular layer for each section using Fiji.

For histological analysis in 8-week-old MMZ- and saline-injected mice, OE sections (3 per antibody per mouse at 25%, 50% and 75% along the anterior-posterior axis) were stained for GAP43 as a marker of immature OSNs (anti-GAP43 primary antibody, 1:1000 for 48 h at 4 °C, NB300-143, Novus Biologicals; donkey anti-rabbit-Alexa Fluor 546 secondary antibody, 1:500 for 1 h at room temperature), or OMP (see above) as a marker of mature OSNs. 2-photon z-stacks of ~1 mm of the dorsal septal OE were collected using the Bergamo system with 2-photon excitation at 800 nm. Image analysis was performed in Fiji. OE width (from the basal lamina to the apical surface) was measured at

3 different locations each on the left and right sides of the OE and averaged for each section. The grand mean width for the three sections per mouse was then calculated. GAP43+ and OMP+ OSNs were counted in each image, and the mean value expressed as linear density (cells per mm) was calculated for each mouse. Widefield fluorescence images of OE turbinates were collected using a Revolve widefield microscope equipped with an Olympus Plan-Apo 20x/0.80NA air objective and Echo software (Echo).

OE sections from mice injected with retrograde tracer were stained with Alexa Fluor 555-conjugated streptavidin to detect biotin-labeled dextran (1:500 in phosphate-buffered saline containing 3% normal donkey serum and 0.3% Triton-X100 for 2 h at room temperature, S32355, Thermo Fisher Scientific), or GAP43 (see above) and Alexa Fluor 555-conjugated streptavidin. In the latter case, sections were first incubated with anti-GAP43 primary antibody (1:1000, 96 h at 4 °C), then with donkey anti-rabbit-Alexa Fluor 647 secondary antibody (1:500) and Alexa Fluor 555-conjugated streptavidin (1:500) in phosphate-buffered saline containing 3% normal donkey serum and 0.3% Triton-X100 for 2 h at room temperature. Confocal z-stacks were collected using an Fluoview 1000 confocal microscope equipped with a Plan Fluorite 40x/1.3NA oil immersion objective and FV 10ASW software (Olympus).

## Statistics

Statistical analyses were performed using Prism 8 or Prism 9 (Graph-Pad). Parametric tests were performed for normally distributed data with equal variances and for $\log_{10}$-transformed data that met these criteria. Welch's ANOVA with Dunnett's T3 multiple comparisons was used for data that were normally distributed but with unequal variance; otherwise, non-parametric tests were performed. Fisher's exact test and Wilcoxon signed rank test were used to compare proportions of responding neurons or glomeruli. Two-tailed unpaired $t$-tests or Mann–Whitney rank sum tests were used to compare two groups, whereas one-way ANOVA with Sidak's multiple comparisons, one-way ANOVA on Ranks with Dunn's multiple comparisons, or two-way ANOVA were used to compare multiple groups. Repeated measures ANOVAs were used where multiple measurements were obtained from the same glomeruli. Friedman's test was used to compare the percentage of responding immature OSNs in the OE across odorants. Nested t-tests and one-way ANOVAs were used to compare parameters obtained from calcium imaging of glomeruli sampled from multiple mice. Linear regression was used to test for relationships between factors. α was 0.05. Exact P values are reported where available. The results of all statistical tests are reported in the figure legends.

## Reporting summary

Further information on research design is available in the Nature Research Reporting Summary linked to this article.

## Data availability

The data that support the findings of this study are available in Zenodo with the identifier https://doi.org/10.5281/zenodo.7154187[102]. The data for relevant figure panels in Figs. 1–7, Fig. S1–S6 and Fig. S8 are provided in the Source Data file. Source data are provided with this paper.

## Code availability

Code for image acquisition and analysis[103], and for electrophysiology analysis[104] is available via Zenodo with the identifiers https://doi.org/10.5281/zenodo.7013674 and https://doi.org/10.5281/zenodo.6993495.

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

## Acknowledgements

This work was supported by grants from the National Institute on Deafness and other Communication Disorders: R03DC014788 and R01DC018516 (CEJC) and the Samuel and Emma Winters Foundation (CEJC), and support from grant 5T32NS007433 (JDG). We would like to thank Alisa Pugacheva for help with preliminary analysis, and Shawn D. Burton and members of the Cheetham lab for helpful discussions.

## Author contributions

Conceptualization, J.S.H., T.K. and C.E.J.C.; Methodology: J.S.H., T.K. and C.E.J.C.; Software: J.S.H, B.L. and C.E.J.C.; Investigation: J.S.H., T.K., A.N.R., T.R.B., J.D.G., B.L., R.J.M., J.T.A. and C.E.J.C.; Formal Analysis: J.S.H., T.K., A.N.R., E.D.W.-B. and C.E.J.C.; Writing- Original Draft: J.S.H. and C.E.J.C.; Writing- Review and Editing: J.S.H., T.K., A.N.R., T.R.B., J.D.G and C.E.J.C.; Visualization: J.S.H. and C.E.J.C.; Supervision: C.E.J.C.; Project Administration: C.E.J.C.; Funding Acquisition: C.E.J.C.

## Competing interests

The authors declare no competing interests.
