## [Peer Review File · Nature Communications]

Immature olfactory sensory neurons provide behaviorally useful sensory input to the olfactory bulbREVIEWER COMMENTS

Reviewer #1 (Remarks to the Author):

In this study combining in vivo 2-photon calcium imaging, slice electrophysiology coupled with optogenetics and behavioral tests, Jane Huang and colleagues examined the contribution of immature olfactory sensory neurons (OSNs) in odor processing. This is an interesting question because OSNs are continuously renewed throughout life. Thus, newly generated OSNs have to integrate a pre-existing network in olfactory bulb glomeruli at the end of a sequential maturation process. However, previous studies have shown that immature OSNs express multiple olfactory receptors, unlike mature OSNs, suggesting that the information they provide may be different and possibly degrade the sensory input to glomeruli.

Here, newly generated immature OSNs vs. mature OSNs were selectively labeled and manipulated using two transgenic driver lines driven by $G\alpha 8$, which is expressed in immature OSNs, and by OMP an established marker for mature OSNs. The conclusion of the paper is that immature OSNs provide behaviorally useful sensory inputs to the olfactory bulb that are as selective as those provided by mature OSNs albeit with perhaps a higher sensitivity (i.e. they are excited by lower concentrations).

Here the maturity / immaturity status of an OSN is defined as whether it express a molecular marker of immaturity ($G\alpha 8$) or a molecular marker of maturity (OMP). This labeling strategy is useful but imperfectly reflects the progressive process of maturation. Obviously, $G\alpha 8$ -expressing OSNs that have just been generated and barely extend axons out of the epithelium are more immature than $G\alpha 8$ -expressing OSNs that have already reached the olfactory bulb and make synapses. These more mature $G\alpha 8$ -expressing OSNs are those studied in the manuscript and the important question is to determine whether their properties are different compared with those of mature OSNs. To my view, together with the previous study of the principal investigator (Cheetham et al., NatureCom 2016), the data suggest that these neurons, while still expressing $G\alpha 8$, are functionally mature and not that different than OMP-expressing OSNs. Thus, I find the title and take home message slightly misleading. In other words, to make the point the authors try to make they should demonstrate that $G\alpha 8$ -expressing OSNs providing sensory inputs within glomeruli are still functionally immature and provide a distinct odor input to the olfactory bulb compared to OMP-expressing mature OSNs. I don't think the data convincingly demonstrate this. Moreover, I have several major concerns with the data and experiments.

2-photon calcium imaging experiments (Figure 1-3)

2-photon calcium imaging experiments were done in P21-23 days old mice, on the dorsal surface of the OB using a panel of 7 odorants. The previous study of the principal author (Cheetham et al., 2016) suggests that in these young mice, glomeruli formed by $G\alpha 8$ -expressing OSNs are more difficult to delimit and less numerous than those formed by OMP-expressing OSNs. This is apparently especially true in the dorsal OB which seems much less innervated by $G\alpha 8$ -expressing OSNs than the ventral OB (as suggested by Fig.4G in the present article). Thus, whereas all glomeruli should be labeled in OMP-GCaMP6s mice, only a fraction may be clearly labeled in $G\alpha 8$ -GCaMP6s mice. There is little information and discussion about this point in the manuscript. It is critical as many results are expressed as % of responding glomeruli. Comparing responses evoked by the same panel of

odorants on a different total number of labeled glomeruli may be misleading. Thus, authors should show and quantify the total number of glomeruli they were able to image in the dorsal OB.

Even if this is done correctly, I guess the authors will compare data that are perhaps not comparable, i.e. different sets and number of glomeruli. What would be ideal is to compare, using dual color imaging, the input provided by G β 8- and by OMP-expressing OSNs innervating the same glomerulus. Admittedly, this may be technically challenging or currently impossible within the time frame of a revision.

From these experiments, the authors conclude that the data provide the first evidence that immature OSNs can detect and transduce odorants binding to generate action potentials that propagate to the OB resulting in calcium influx at glomerular axon terminals (line 142). I think the data show that odorants evoke calcium responses in axon terminals of G β 8-GCaMP6s OSNs suggesting that immature OSNs can detect and transduce odorants binding. The possibility of an indirect activation through ephaptic coupling (Bokil et al., 2001) cannot be excluded.

The authors conclude that immature OSNs have a lower concentration threshold than mature OSNs but are less effective in encoding the concentration of an odorant (line 262). This is a strong statement for a rather basic analysis using only 7 odorants applied at two concentrations. Moreover, the only feature that was analyzed was the amplitude of the response. The time course of the response is also important and may encode odor concentration.

Patch-clamp experiments in OB slices (Figure 4)

The objective was to determine which OB neurons receive direct monosynaptic input from immature OSNs (line 271). Thus, the authors made whole-cell recordings of external tufted cells (ETC) and mitral cells (MC) in transgenic mice expressing the channelrhodopsin variant ChIEF in OMP-expressing OSNs or in G β 8-expressing OSNs. Experiments were done in voltage-clamp at $V_h = -70$ mV, using a KCl based internal solution. OSNs were stimulated using a 50 ms-long light pulse. This induced inward currents in some but not all mitral and ET cells. I have several serious concerns about these experiments.

Why ETC and MC? why not other types of interneurons or other types of tufted cells? Please justify and also indicate how did you identified/defined an ETC? The literature is still confusing on this question but, at least in the articles they cite (Hayar et al., 2004; Bourne and Schoppa 2017, Gire et al., 2012; Najac et al., 2011; Vaaga and Westbrook, 2016), external tufted cells (ETC) refer to a specific population of juxtglomerular neurons with unique morphological (unlike other tufted cells they do not have lateral dendrites) and intrinsic membrane properties (they are spontaneously bursting) and an important role in driving excitation of the glomerular network. Thus, ETCs are different from other tufted cells with cell bodies also found in or near the glomerular layer but with morphologies and membrane properties that are more similar to those of mitral or middle tufted cells (see these recent papers for comparison: Jones et al., *Front. Cell. Neurosci.* (2020) 14:614377, or Sun et al., *JNeurosci* (2020) Aug 5;40(32):6189-6206).

Why a 50 ms long light stimulus? A monosynaptic glutamatergic input is usually defined as a fast rising AMPA-mediated EPSC that occurs within a short ($\approx 1-2$ ms) and low trial-to-trial variability delay after the stimulation of the presynaptic axons. This is why it is preferable to stimulate the presynaptic axons with a short stimulus inducing a synchronized input. A 50 ms long optical

stimulation likely evokes a barrage of action potentials in the presynaptic axons and unsynchronized glutamatergic inputs onto the postsynaptic cells making difficult to detect a monosynaptic input.

Instead of analyzing the kinetics of the evoked currents, the authors used 4-AP and TTX to block polysynaptic contributions and favor monosynaptic responses. However, it is not clear if this was done systematically and whether all the cells/amplitudes included in the data set in Fig.4D-F were obtained in the presence of TTX and 4-AP. Line 673 in the methods (Sequential recordings in normal ACSF and in the presence of TTX and 4-AP were performed for a subset of cells) suggests it was not. Then, polysynaptic responses, possibly GABAergic (see next point), may have been included in the data set. Moreover, it is not clear whether this treatment entirely blocked polysynaptic responses. Pluri-synaptic dendro-dendritic interactions systematically contribute and prolong the MC response. The duration of the MC response shown in red in Figure 4C suggests that this pluri-synaptic component persisted in TTX and 4-AP.

Why using an internal solution containing KCl? With this internal, at a holding potential of -70 mV, inhibitory GABAergic currents contribute to the evoked inward current. The fact that this response was blocked by NBQX and d-AP5 in a subset of cell does not prove that the responses were glutamatergic as an inhibitory di-synaptic response would also be blocked.

A cell was classified as having a monosynaptic response if the EPSC peak amplitude exceeded 3 SD of the pre-stimulus baseline (line 680 in the methods). This is wrong. A polysynaptic response could also exceed a 3 SD pre-stimulus baseline.

Only a subset of mitral cell responded to the stimulation of mature OMP-Chief-Citrine consistent with the weaker monosynaptic input to MC vs tufted cells reported previously (line 284). This is not the reason. Any intact mitral cell should respond to the stimulation in OMP-ChIEF mice, including in the presence of TTX and 4-AP, assuming that OSNs are effectively stimulated. I guess non responsive MC most likely had severed apical dendrites (this often happens in slices). Was this checked?

Cells were only included in the analysis if their holding current at -70 mV was less than -300 pA (line 665). This is not a good idea. Intact MC have a low input resistance (<100 M Ω) and thus necessitate a large holding current to clamp them at -70 mV. Using this cut-off criterion, the authors may have preferentially selected MC with severed apical dendrites (and thus no response).

MMZ experiments (Figure 5 and 6)

In these experiments the authors ablated OSNs with MMZ. Then they tested olfactory sensitivity 3-7 days after this treatment assuming that any olfactory detection is mediated by newly generated $\text{G}\alpha\text{8}$ -expressing OSNs. They found that mice can detect odor as early as 5 days post MMZ.

Liberia et al. (2019) recently showed that newly generated OSN axons reach the OB and robustly innervate specific glomeruli ~10 days following basal cell division. Here the behavioral tests were

done 3 to 7 days after OSNs were ablated by MMZ. Thus, to strengthen their conclusion, it is important that the authors demonstrate that G α 8-expressing OSNs reach the OB and make synapses within glomeruli earlier than previously shown and as early as 5 days after their birth.

Reviewer #2 (Remarks to the Author):

In the manuscript by Huang et al., the authors implement diverse genetic, imaging, electrophysiological, and behavioral tools to thoroughly investigate the tuning and connectivity of immature OSNs, alongside the sufficiency of immature OSNs to influence olfactory guided behavior. The main findings that immature OSNs are odor selective and sufficient for detection and discrimination behaviors are novel and broadly interesting in the context of sensory processing. However, enthusiasm is somewhat tempered by a few key issues associated with analysis methods and interpretation. Additionally, the strength of the conclusions drawn from the behavioral data could be improved with an additional imaging and/or electrophysiology experimentation. Overall this is a good study that deserves further consideration with attention to the relevant points mentioned below.

Major points:

1. Comparing dF/F amplitudes across genotypes with different expression levels confounds the main findings Fig 1F and the concentration analysis in Fig3 E-H. dF/F shouldn't be used as a measure of relative strength of the response compared across genotypes with different GCaMP expression systems. Also, it is unclear how Ca²⁺ in an immature axon terminal correlates with firing or neurotransmitter release onto targets. This should at least be addressed in some capacity.
2. The measure of % responsive glomeruli is dependent on how many glomeruli are outlined. For this to be objective and unbiased by genotype it would have to be done using a secondary marker for glomeruli. Lifetime sparseness is a good way around this confound (i.e. specifically picking responsive glomeruli and determining how many odors they respond to) but this analysis is only applied in Figure 2. The analysis throughout would be more convincing if "% glomeruli responding" measures were replaced with lifetime sparseness.
3. Regarding the slice ephys experiments: If the purpose is to demonstrate that the imaging represents physiologically meaningful input from immature OSN axons onto OB neurons, it seems important to get recordings from the same part of the OB that is used for the imaging comparisons. This is especially true given that (as noted in the methods) immature OSN axons more robustly innervate the ventral OB.
4. Detection and habituation behavioral experiments look underpowered based on the variability of the post-MMZ conditions. The main claim is that immature OSNs rescue olfactory guided behavior at 5+ days post MMZ based on the lack of a significant difference between saline and MMZ at 5+ days. However, each behavior still seems substantially impaired at 5 days, but fails to reach significance because of the noisiness of the behavioral assays. For example, the claim that there is no improvement with immature OSN integration could just as easily be made from the data shown in

Fig6B, F, G, and H considering that there is no significant difference between 3d post MMZ and 5, 6, or 7d post MMZ.

5. Considering the claims that are hinging on behavior post-MMZ (in the context of no mature OSNs) it would be useful to consider (1) imaging from immature axons after MMZ (showing activity in the OB), and/or (2) slice ephys after MMZ showing that monosynaptic connectivity is restored by immature OSNs after 5+ days.

Minor points:

1. Why in some cases is the median shown with individual values, while in other cases box and whisker plots (but no individual data) is shown? (Fig1 F vs G) Does this imply that in Fig1 G all responding glomeruli are treated as independent? Responding glomeruli should be nested within animals to account for dependence within animals (window quality, location, overall expression, etc.)

2. It would be useful to see example images of the staining in the MMZ conditions used to count immature and mature OSNs.

3. Typo in line 233 (partial sentence duplicated).

4. Detection behavior in which mice max out at an arbitrary time should not be analyzed as a continuous variable (i.e. in the fitting in Fig 6C). There actually does appear to be a negative correlation between digging time and time to find the buried food that is masked by all the arbitrary "10 min" data points.

5. Please show error bars in Fig 6F.

Reviewer #3 (Remarks to the Author):

The manuscript by Huang et al provides important new insights into the contribution of immature olfactory sensory neurons to functional signaling in the olfactory bulb. The experiments are, in general, well executed. This manuscript does provide one of the most comprehensive examinations of the dynamics and contributions of the regenerative process in olfactory signaling since it combines morphologic, electrophysiologic, and behavioral approaches. There are a few areas of concern regarding this manuscript. Specifically, in several places, the authors appear to misstate what is actually being measured, carefully evaluate the limits of their observations, or obscure some technical complexities that may confuse the average reader.

In figure 1b, it is not possible to assess the potential overlap in the transition from Ggamma8 to OMP in a merged image. Although they state that there is only 3% co-labeling that is likely highly dependent on the thresholds that they use in a presumably subjective counting experiment. In figure 1D the Ggamma8 positive glomeruli look very different than the OMP positive ones. I am not convinced that the Ggamma8-GCAMP positive fibers enter into glomeruli. This is a critical since it is

the only image that shows the OB glomeruli (other than the low mag Figure 4G that raises similar concerns) yet this forms the basis for data in several subsequent figures and conclusions.

The argument about the tuning of immature and mature OSNs based on their putative expression of additional receptors is not particularly informative. If one were to suggest that mature OSNs express exactly 1 OR (that determines the target glomeruli where it coalesces) and immature OSNs express two receptors (one corresponding to the cognate OR and a random one from the entire repertoire) the glomerular-level response to individual odors would look like the response profile of the common OR.

The authors describe the experiments in figure 6DE as an odor preference when it should be characterized as a food detection test. Mineral oil is usually used as a null carrier and contributes no odor that mice could choose between.

The authors are acknowledged for providing impressive, quantitative data at the septum on the efficiency of their MMZ treatment. However, MMZ is highly regional in its efficacy – considerably more efficient at the septum than in lateral and ventral areas. Either a full coronal section OMP stain or at least a statement about overall efficiency is essential to properly interpret the behavior data post-MMZ.

Reviewer 1

We were pleased that the Reviewer thought that our study addressed an interesting question. We have carefully considered their comments and have significantly revised the manuscript, including adding a large amount of additional data, to address their concerns. A detailed point-by-point response is included below.

General comments

To my view, together with the previous study of the principal investigator (Cheetham et al., NatureCom 2016), the data suggest that these neurons, while still expressing G γ 8, are functionally mature and not that different than OMP- expressing OSNs. Thus, I find the title and take home message slightly misleading. In other words, to make the point the authors try to make they should demonstrate that G γ 8-expressing OSNs providing sensory inputs within glomeruli are still functionally immature and provide a distinct odor input to the olfactory bulb compared to OMP-expressing mature OSNs. I don't think the data convincingly demonstrate this.

We think that our finding that G γ 8-expressing OSNs provide sensory input to the OB is important, and that the relevance of this finding is not negated by the fact that some properties of this input are similar to those of mature OSNs. We have followed the Reviewer's suggestion below to considerably strengthen our data set addressing odorant concentration, finding that both the latency and amplitude of inputs from immature OSNs provide information about odorant concentration that is not available from mature OSN inputs. Hence, our data suggest that immature and mature OSNs provide functionally distinct odor information to glomeruli.

We have edited the Discussion (lines 742-761) to clarify the importance of our findings, highlighting both differences in immature OSN input identified in this study, and our previous finding that immature OSNs undergo a higher level of presynaptic terminal formation and elimination than their mature counterparts (Cheetham et al., Nature Comms 7:10729 (2016)). We also note that immature OSNs may play a particularly important role during development and regeneration of OB input. Overall, we believe that our data support the conclusions that immature OSNs provide a functionally distinct information stream to OB neurons and play a previously unappreciated role in olfactory-guided behavior.

2-photon calcium imaging experiments (Figure 1-3)

The previous study of the principal author (Cheetham et al., 2016) suggests that in these young mice, glomeruli formed by G γ 8-expressing OSNs are more difficult to delimit and less numerous than those formed by OMP-expressing OSNs. This is apparently especially true in the dorsal OB which seems much less innervated by G γ 8-expressing OSNs than the ventral OB (as suggested by Fig.4G in the present article). Thus, whereas all glomeruli should be labeled in OMP-GCaMP6s mice, only a fraction may be clearly labeled in G γ 8-GCaMP6s mice. There is little information and discussion about this point in the manuscript. It is critical as many results are

expressed as % of responding glomeruli. Comparing responses evoked by the same panel of odorants on a different total number of labeled glomeruli may be misleading. Thus, authors should show and quantify the total number of glomeruli they were able to image in the dorsal OB.

We had not adequately addressed the important issue of detection of glomeruli in G γ 8-GCaMP6s mice in the previous version of the manuscript. We have now performed additional analysis to quantify the density of detected glomeruli in 3-week-old mice from four lines: G γ 8-GCaMP6s, OMP-GCaMP6s, G γ 8-sypGFP-tdT and OMP-sypGFP-tdT. This enabled us both to determine whether fewer glomeruli were detectable in mouse lines expressing a G γ 8-tTA-driven vs. an OMP-tTA-driven reporter; and whether fewer glomeruli were detectable when the reporter was GCaMP6s, which can have low resting fluorescence, vs. lines expressing both cytosolic tdTomato and GFP-tagged-synaptophysin. We found no effect of either OSN maturity or the reporter protein(s) expressed on the density of glomeruli that could be identified (Fig. 2B, Results lines 143-161). Hence, we concluded that we are able to identify the majority of glomeruli within a field of view in both the G γ 8-GCaMP6s and OMP-GCaMP6s mouse lines at 3 weeks of age, i.e., the fact that fewer G γ 8-expressing axons than OMP-expressing axons enter many glomeruli does not preclude glomerular detection.

Even if this is done correctly, I guess the authors will compare data that are perhaps not comparable, i.e. different sets and number of glomeruli. What would be ideal is to compare, using dual color imaging, the input provided by G γ 8- and by OMP-expressing OSNs innervating the same glomerulus. Admittedly, this may be technically challenging or currently impossible within the time frame of a revision.

We carefully considered, both while performing our original experiments and in response to the Reviewer's comments, the possibility of performing dual color calcium imaging in the same glomeruli, e.g., using GCaMP6s in G γ 8-expressing OSNs and jRGECO1a (or similar) in OMP-expressing OSNs. However, we concluded that this approach would not give us a genuine comparative measurement due to the necessity to employ two different genetically encoded calcium indicators (GECIs), as noted in Dana et al. (eLife 5 e12727 (2016)). First, the maximal fluorescence changes elicited by calcium binding are 3-4-fold smaller for the best currently available red GECIs than for GCaMP6s. Second, the signal to noise ratio is lower for red GECIs due to their relatively low 2-photon absorption cross section. Third, lysosomal accumulation of jRGECO1a, the most sensitive red GECI, further reduces fluorescence changes. Fourth, red GECIs form multiple species, some of which generate green fluorescence but do not report calcium.

A further technical challenge would be how to express the red GECI in mature OSNs. To our knowledge, there is no commercially available purely Cre-dependent red GECI reporter mouse line (we note that there is a Cre/tTA dependent RCaMP1 line, but this is unsuitable here as the only way to drive GCaMP6s expression specifically in G γ 8-expressing OSNs is using tTA). Hence, we would need to employ a viral strategy for red GECI expression in mature OSNs, adding

additional technical complexity and further confounding interpretation as viral expression can result in higher expression levels, cytotoxicity and altered kinetics (Dana et al., PLoS One 9 e108697 (2014)).

However, this is a very important point, so we designed a different experimental strategy to address it. Via several generations of breeding, we produced $\text{G}\gamma 8\text{-GCaMP6s}$ and OMP-GCaMP6s mice that were also homozygous for M72-RFP (Fig. 4C,D; note that homozygosity was necessary to visualize the M72 glomerulus *in vivo*; as employed in Arneodo et al. (Nature Comms 9 1347 (2018))). This enabled us to compare odor responses of immature and mature OSN axons in the lateral M72 glomerulus and a field of view centered around that glomerulus. We found no difference in immature vs. mature OSN odorant responses in lateral M72 glomeruli in terms of the number of odorants to which they responded or the lifetime sparseness of odorant responses (Fig. 4E,F). We also found no differences in the number of odorants responded to or lifetime sparseness in glomeruli surrounding the lateral M72 glomerulus between $\text{G}\gamma 8\text{-GCaMP6s-M72-RFP}$ and $\text{OMP-GCaMP6s-M72-RFP}$ mice (Figure 4G,H). These new data are described in the Results (lines 319-335) and strengthen our conclusion that odorant selectivity is similar for immature and mature OSN axons *in vivo*.

From these experiments, the authors conclude that the data provide the first evidence that immature OSNs can detect and transduce odorants binding to generate action potentials that propagate to the OB resulting in calcium influx at glomerular axon terminals (line 142). I think the data show that odorants evoke calcium responses in axon terminals of $\text{G}\gamma 8\text{-GCaMP6s}$ OSNs suggesting that immature OSNs can detect and transduce odorants binding. The possibility of an indirect activation through ephaptic coupling (Bokil et al., 2001) cannot be excluded.

We have revised the results text (now lines 177-179) as suggested by the Reviewer, which now reads “... the data in Fig. 2 provide the first evidence that odorants evoke calcium responses in immature OSN axon terminals, suggesting that immature OSNs can detect and transduce odorant binding”. Similarly, we revised the Discussion text (lines 584-586): “Furthermore, our *in vivo* and *ex vivo* calcium imaging data demonstrate odor-evoked increases in intracellular calcium in both the somata and axon terminals of immature OSNs (Fig. 2), suggesting that immature OSNs can detect and transduce odorant binding”.

We have also added new data showing that immature OSNs in the olfactory epithelium respond to odors using *ex vivo* GCaMP6s imaging in a hemi-head preparation (Fig. 2F-H, Results lines 171-179). These new data make it very unlikely that ephaptic transmission is solely responsible for odor-evoked calcium responses in immature OSN presynaptic terminals in the OB, but we cannot rule out some potential contribution. Therefore, we have also added discussion of the possible contribution of ephaptic transmission (lines 597-604).

The authors conclude that immature OSNs have a lower concentration threshold than mature OSNs but are less effective in encoding the concentration of an odorant (line 262). This is a strong statement for a rather basic analysis using only 7 odorants applied at two concentrations. Moreover, the only feature that was analyzed was the amplitude of the

response. The time course of the response is also important and may encode odor concentration.

We agree with the Reviewer that our original data set using only two concentrations needed to be improved upon. Therefore, we performed a new set of *in vivo* imaging experiments in 7 OMP-GCaMP6s and 7 G γ 8-GCaMP6s mice to address this question. We stimulated mice with four concentrations each of five known dorsal surface activating odorants (a total of 21 stimuli including the blank was the maximum possible by extending our existing olfactometer, given pandemic-related supply chain constraints). We thank the Reviewer for the suggestion to analyze response time course and have added analysis of response latency and time to peak, which yielded interesting data.

Our new data set is in much better agreement with previous studies, with both the number of responsive glomeruli and the amplitude of glomerular responses increasing across the tested concentration range in OMP-GCaMP6s mice. We found that glomeruli in G γ 8-GCaMP6s mice also encode concentration and provide concentration information that is not available from mature OSN inputs. Specifically, both the amplitude and latency of responses provided information across the entire concentration range that we tested (0.5 – 10 %) in G γ 8-GCaMP6s mice, whereas amplitude and latency were only significantly different between the lowest pair of odorant concentrations tested (0.5 % and 1 %) in OMP-GCaMP6s mice. These new data are shown in the new Fig. 3 and described in the Results (lines 217-257).

A more extensive analysis of concentration coding by immature OSNs would be interesting to perform but we believe is outside the scope of the current study. However, we agree with the caveats noted by the Reviewer and have emphasized them in both the Results (line 255) and the Discussion (lines 664-679).

Patch-clamp experiments in OB slices (Figure 4)

Why ETC and MC? Why not other types of interneurons or other types of tufted cells? Please justify and also indicate how did you identified/defined an ETC? The literature is still confusing on this question but, at least in the articles they cite (Hayar et al., 2004; Bourne and Schoppa 2017, Gire et al., 2012; Najac et al., 2011; Vaaga and Westbrook, 2016), external tufted cells (ETC) refer to a specific population of juxtglomerular neurons with unique morphological (unlike other tufted cells they do not have lateral dendrites) and intrinsic membrane properties (they are spontaneously bursting) and an important role in driving excitation of the glomerular network. Thus, ETCs are different from other tufted cells with cell bodies also found in or near the glomerular layer but with morphologies and membrane properties that are more similar to those of mitral or middle tufted cells (see these recent papers for comparison: Jones et al., *Front. Cell. Neurosci.* (2020) 14:614377, or Sun et al., *Jneurosci* (2020) Aug 5;40(32):6189-6206).

We apologize that we had not clearly explained the goal of these experiments in the previous version of the manuscript. While it would be very interesting to determine all of the excitatory and inhibitory neuron types that receive input from immature OSNs, such data would comprise

a separate study in its own right. Our goal here was rather to determine whether immature OSNs provide monosynaptic input to OB neurons, as we could not ascertain this from our previous study (Cheetham et al., *Nature Comms* 7 10729 (2016)), which employed extracellular *in vivo* recordings.

We performed a new set of slice electrophysiology experiments in response to Reviewer 1's other comments (see below). We decided to focus on a single cell type, STCs, which are known to receive monosynaptic input from mature OSN axons (Sun et al., *J. Neurosci.* 40 6189 (2020)). STCs are less specialized than external TCs and more closely resemble the classic category of TCs that function as OB output neurons. STCs, however, are much easier to identify than the classic middle TCs (mTCs) that are scattered throughout the EPL, and their properties are more homogenous than those of mTCs, which vary depending on the depth in the EPL at which they reside (Shepherd et al., in *The Synaptic Organization of the Brain* 165 (2004); Igarishi et al., *J. Neurosci.* 32 7970 (2012); Nagayama et al., *Front. Neur. Circuits* 8:98 (2014)). Another practical consideration is that STC apical dendrites are more likely to be intact following slicing than those of deeper TCs.

We employed a rigorous set of criteria based on published data (Antal et al., *Eur. J. Neurosci.* 24 1124 (2006); De Saint Jan et al., *J. Neurosci.* 29 2043 (2009); Liu et al., *J. Neurophysiol.* 107 473 (2011); Sun et al., *J. Neurosci.* 40 6189 (2020)) to identify STCs, using both morphological (laminar location, presence of a lateral dendrite in the EPL in addition to the apical dendrite) and functional (non-bursting spike patterns without depolarizing envelope) properties. Fig. 5D and 5E have been added to demonstrate examples of these properties.

Why a 50 ms long light stimulus? A monosynaptic glutamatergic input is usually defined as a fast rising AMPA-mediated EPSC that occurs within a short ($\approx 1-2$ ms) and low trial-to-trial variability delay after the stimulation of the presynaptic axons. This is why it is preferable to stimulate the presynaptic axons with a short stimulus inducing a synchronized input. A 50 ms long optical stimulation likely evokes a barrage of action potentials in the presynaptic axons and unsynchronized glutamatergic inputs onto the postsynaptic cells making difficult to detect a monosynaptic input.

We thank the reviewer for the suggestion to use a short light stimulus. For the revised experiments, we first established a power curve measuring the photoactivated current in response to increasing light duration and intensity, with APV present to isolate AMPA-mediated currents. Each set of stimulus trials included 10 sweeps of light stimuli at 0.25, 1, or 2 ms duration at increasing light intensities (10, 20, 50, 80, or 100%). We have added Fig. S3 as an example of a complete power curve recorded from one STC.

For analysis, we selected the shortest light duration and intensity parameter (1 ms at 100 %) that consistently evoked a photoactivated current in the recorded cells. The resulting EPSCs showed onset latencies of less than 2 ms and had low trial-to-trial jitter. The onset latencies and jitter were comparable for photostimulation of immature G γ 8-ChIEF-Citrine and mature OMP-

ChIEF-Citrine OSN axons (Fig. 5K,L) and were consistent with previously reported monosynaptic OSN input to STCs (Sun et al., J. Neurosci. 40 6189 (2020)).

Instead of analyzing the kinetics of the evoked currents, the authors used 4-AP and TTX to block polysynaptic contributions and favor monosynaptic responses. However, it is not clear if this was done systematically and whether all the cells/amplitudes included in the data set in Fig.4D-F were obtained in the presence of TTX and 4-AP. Line 673 in the methods (Sequential recordings in normal ACSF and in the presence of TTX and 4-AP were performed for a subset of cells) suggests it was not. Then, polysynaptic responses, possibly GABAergic (see next point), may have been included in the data set. Moreover, it is not clear whether this treatment entirely blocked polysynaptic responses. Pluri-synaptic dendro-dendritic interactions systematically contribute and prolong the MC response. The duration of the MC response shown in red in Figure 4C suggests that this pluri-synaptic component persisted in TTX and 4-AP.

We agree with the reviewer that using only pharmacological manipulation instead of kinetic analysis was not sufficiently rigorous in isolating and determining the presence of monosynaptic responses. As described above, in our new experiments, we decided to forego the use of 4-AP and TTX and instead use short duration 1 ms light pulses to isolate monosynaptic responses.

We note that some STCs show a prolonged component in the photoactivated current (see example traces in Fig. 5F,G), as reported previously for OSN inputs to STCs (Sun et al., J. Neurosci. 40 6189 (2020); Jones et al., Front. Cell. Neurosci. 14:614377 (2020)), that is likely due to feedforward excitation. This feature is present in both Gy8-ChIEF-Citrine and OMP-ChIEF-Citrine mice.

Why using an internal solution containing KCl? With this internal, at a holding potential of -70 mV, inhibitory GABAergic currents contribute to the evoked inward current. The fact that this response was blocked by NBQX and d-AP5 in a subset of cell does not prove that the responses were glutamatergic as an inhibitory di-synaptic response would also be blocked.

The reviewer is correct that a KCl internal is suboptimal for isolating glutamatergic excitatory currents. In the revised set of experiments, we instead used a potassium gluconate internal, which enabled both current clamp recordings of STC firing patterns and voltage clamp recordings of excitatory photoactivated input. Holding the cell at -70 mV with APV in the bath allowed us to conclude that we had isolated AMPA-mediated, glutamatergic currents (Fig. 5F-G).

A cell was classified as having a monosynaptic response if the EPSC peak amplitude exceeded 3 SD of the pre-stimulus baseline (line 680 in the methods). This is wrong. A polysynaptic response could also exceed a 3 SD pre-stimulus baseline.

We had worded this statement poorly in the previous version of the manuscript. In response to the reviewer's comment, we have revised the Methods section (lines 925-928) to read: "A cell was classified as having a response if the EPSC peak amplitude exceeded three standard deviations of the pre-stimulus baseline current. Only STCs showing responses with an onset latency shorter than 2 ms were defined as receiving monosynaptic input from OSN axons".

Only a subset of mitral cell responded to the stimulation of mature OMP-Chief-Citrine consistent with the weaker monosynaptic input to MC vs tufted cells reported previously (line 284). This is not the reason. Any intact mitral cell should respond to the stimulation in OMP-ChIEF mice, including in the presence of TTX and 4-AP, assuming that OSNs are effectively stimulated. I guess non responsive MC most likely had severed apical dendrites (this often happens in slices). Was this checked?

We agree that an intact apical dendrite may be correlated with the responsiveness of a cell. In our revised experiments, we filled every recorded cell with AF594 to visualize cell morphology on the rig microscope and noted whether we were able to see the apical tuft or an intact apical dendrite. We then correlated the presence or absence of an apical tuft/dendrite with whether the cell showed a response. In both Gy8-ChIEF-Citrine and OMP-ChIEF-Citrine mice, there are responsive cells in which we could not visualize an apical dendrite, and there are also cells that had apical dendrites but no responses. The inability to visualize apical dendrites/tufts in some responsive cells may be explained by their apical dendrites running deep within the slice. Indeed, we observed oscillatory resting membrane potentials in many of our cells, which is correlated with intact apical dendrites (Carlson et al., J. Neurosci. 20 211 (2000); Schoppa and Westbrook, Neuron 31 639 (2001); Hayar et al., J. Neurosci. 25 8197 2005).

The presence of an apical dendrite did not guarantee monosynaptic input from photoactivated OSNs. The lack of monosynaptic input onto some recorded cells may be explained by the subtypes of STCs we recorded from. For example, vasopressin-expressing STCs do not show excitatory input following electrical olfactory nerve stimulation (Lukas et al. eNeuro 6:ENEURO.0431-18.2019 (2019)). Although we did not distinguish between possible subtypes of STCs included in our dataset, it is plausible that some non-responsive cells belonged to a physiologically distinct group from the responsive cells.

Cells were only included in the analysis if their holding current at -70 mV was less than -300 pA (line 665). This is not a good idea. Intact MC have a low input resistance (<100 MΩ) and thus

necessitate a large holding current to clamp them at -70 mV. Using this cut-off criterion, the authors may have preferentially selected MC with severed apical dendrites (and thus no response).

In our new set of experiments, we used a potassium gluconate internal. We have revised our exclusion criteria accordingly (Methods, lines 900-901): *“Cells were excluded from analysis if their resting membrane potential was depolarized above -45 mV.”*.

MMZ experiments (Figure 5 and 6)

In these experiments the authors ablated OSNs with MMZ. Then they tested olfactory sensitivity 3-7 days after this treatment assuming that any olfactory detection is mediated by newly generated G γ 8-expressing OSNs. They found that mice can detect odor as early as 5 days post MMZ. Liberia et al. (2019) recently showed that newly generated OSN axons reach the OB and robustly innervate specific glomeruli ~10 days following basal cell division. Here the behavioral tests were done 3 to 7 days after OSNs were ablated by MMZ. Thus, to strengthen their conclusion, it is important that the authors demonstrate that G γ 8-expressing OSNs reach the OB and make synapses within glomeruli earlier than previously shown and as early as 5 days after their birth.

We agree that this is an important point. We attempted several approaches to answer this question, but found that the persistence of a high density of degenerating immature OSN axons at 5-6 days post-MMZ, which has been described previously (e.g. Fig. S2 in Tsai & Barnea Science 344 197 (2014); Fig. 4 in Kikuta et al., J. Neurosci. 36 2657 (2015); Fig. 1,2 in Blanco-Hernandez et al., PloS One 7:e46338 (2012)) posed major technical challenges.

Importantly, however, we were able to employ retrograde tracer injections to demonstrate that the axons of newly generated immature OSNs reach the OB as early as 5 days post-MMZ. We injected biotin-labeled dextran, a retrograde tracer, into the anterior dorsal OB of mice that had received MMZ 5 days earlier. This method has been used previously to identify OSNs that project into glomeruli (Rodriguez-Gil et al., PNAS 112 5821 (2015)). We identified sparse, tracer-labeled immature OSNs in the OE when mice were perfused 24 hours later (Fig. 6H). These data are described in the Results (lines 468-474) and Discussion (lines 706-709). This finding strengthens our data demonstrating that mice begin to detect and discriminate odors as early as 5 days post-MMZ, at a time point when no mature OSNs are present in the OE (Fig. 6A-G).

We have also added discussion (lines 709-713) of our data in the context of Liberia et al. (2019), which reported robust glomerular innervation by 10-day-old OSNs, as well as Rodriguez-Gil et al. (PNAS 112 5821 (2015)), which reported strong glomerular innervation by 8-day-old OSNs. We note that neither of these studies specifically focused on when the very first axons enter the OB, and hence did not perform a comprehensive analysis of glomeruli throughout the OB. Nevertheless, in their images, very sparse OSN axons are visible in the glomerular layer as early as 5 days post-tamoxifen injection in *Ascl1-creER* mice (Liberia et al. Fig. 8H [5 days] and Fig. 10A [7 days]; Rodriguez-Gil et al. Fig. 5J).

Reviewer 2

We were pleased that the Reviewer thought that our findings are novel and broadly interesting. We have significantly revised the manuscript to address their specific concerns, as detailed below.

Major points

1. Comparing dF/F amplitudes across genotypes with different expression levels confounds the main findings Fig 1F and the concentration analysis in Fig3 E-H. dF/F shouldn't be used as a measure of relative strength of the response compared across genotypes with different GcaMP expression systems. Also, it is unclear how Ca^{2+} in an immature axon terminal correlates with firing or neurotransmitter release onto targets. This should at least be addressed in some capacity.

We employed $\Delta F/F$ because it is a very widely used measure of response amplitude for analysis of odor-evoked calcium responses, as well as more broadly for neuronal calcium imaging. Our rationale was that $\Delta F/F$ is independent of the expression level of the calcium indicator because the odor-evoked change in fluorescence is normalized to the resting fluorescence (Maravall et al., *Biophys. J.* 78 2655 (2000)), and $\Delta F/F$ has been used previously to compare responses in neurons expressing different generations of GcaMP6 reporters, which are known to have markedly different expression levels (Daigle et al., *Cell* 174 465-480 (2018)), and to compare the performance of different genetically encoded calcium indicators (Chen et al., *Nature* 499 295 (2013)).

However, we agree that we do not know the expression levels of GCaMP6s in immature vs. mature OSN axons. In addition, after further consideration, we decided that the incomplete innervation of glomeruli by immature OSN axons (e.g., Kim & Greer 2000; see Results lines 144-147) could confound a $\Delta F/F$ -based comparison across genotypes. Therefore, we now use $\Delta F/F$ to compare response amplitudes only within genotypes (across odors in Fig. 2D,E and across concentrations in Fig. 3F,G)

We have also added discussion on the important relationships between calcium concentration, firing and neurotransmitter release to the manuscript (lines 606-629). To avoid over-interpretation, we have also edited the main conclusion that we draw from the *in vivo* calcium imaging data and the new *ex vivo* calcium imaging data (both shown in Fig. 2) to remove any reference to action potentials. These sections now read: Results (lines 177-179): *"Taken together, the data in Fig. 2 provide the first evidence that odorants evoke calcium responses in immature OSN axon terminals, suggesting that immature OSNs can detect and transduce odorant binding"* and Discussion (lines 584-586): *"Furthermore, our in vivo and ex vivo calcium imaging data demonstrate odor-evoked increases in intracellular calcium in both the somata and axon terminals of immature OSNs (Fig. 2), suggesting that immature OSNs can detect and transduce odorant binding"*.

Attempting to determine these relationships would be beyond the scope of the current study. Indeed, the relationship between action potential firing and presynaptic terminal calcium concentration has not been determined for mature OSNs, likely due to the technical challenges associated with simultaneous recording of action potentials in OSN somata in the nose and calcium responses in their axon terminals in the OB. Considerably more is known about the relationship between mature OSN presynaptic calcium concentration and neurotransmitter release (see Discussion, lines 618-629). This relationship could theoretically be directly determined for immature OSNs by performing simultaneous imaging of synaptopHluorin and a red genetically encoded calcium indicator (GECI). However, to our knowledge, there is no available purely Tet-dependent red GECI line, meaning that this experiment is not currently technically feasible.

2. The measure of % responsive glomeruli is dependent on how many glomeruli are outlined. For this to be objective and unbiased by genotype it would have to be done using a secondary marker for glomeruli. Lifetime sparseness is a good way around this confound (i.e. specifically picking responsive glomeruli and determining how many odors they respond to) but this analysis is only applied in Figure 2. The analysis throughout would be more convincing if “% glomeruli responding” measures were replaced with lifetime sparseness.

We agree that lifetime sparseness provides important information about the number of odors to which odor-responsive glomeruli respond as well as the relative strength of those responses. Therefore, we have included lifetime sparseness analysis for all GCaMP6s imaging data in the manuscript: we have performed lifetime sparseness analysis of odorant responses in the lateral M72 glomerulus (Fig. 4F) and surrounding glomeruli (Fig. 4H; see response to Reviewer 1 for rationale for adding these data); and added lifetime sparseness data to the updated analysis of the concentration dependence of odor responses (Fig. 4I). We believe that the additional lifetime sparseness analysis considerably strengthens our conclusion that odor selectivity is similar for immature and mature OSNs.

We also agree that we had not adequately addressed the important issue of detection of glomeruli in G γ 8-GCaMP6s mice in the previous version of the manuscript. We performed additional analysis to quantify the density of detected glomeruli in 3-week-old mice from four lines: G γ 8-GCaMP6s, OMP-GCaMP6s, G γ 8-sypGFP-tdT and OMP-sypGFP-tdT. This enabled us both to determine whether fewer glomeruli were detectable in mouse lines expressing a G γ 8-tTA-driven vs. an OMP-tTA-driven reporter; and whether fewer glomeruli were detectable when the reporter was GCaMP6s, which can have low resting fluorescence, vs. lines expressing both cytosolic tdTomato and GFP-tagged-synaptophysin, for which fluorescence is not activity dependent. We found no effect of either OSN maturity or the reporter protein(s) expressed on the density of glomeruli that could be identified (Fig. 2B, Results lines 150-161). Hence, we concluded that we are able to identify the majority of glomeruli within a field of view in both the G γ 8-GCaMP6s and OMP-GCaMP6s mouse lines at 3 weeks of age, i.e., the fact that fewer G γ 8-expressing axons than OMP-expressing axons enter many glomeruli does not preclude glomerular detection. We have therefore retained our analysis of percentage responsive glomeruli, which we believe is important in showing that many glomeruli receive odor input

from immature OSNs and provides information that is complementary to the analysis of lifetime sparseness described above.

3. Regarding the slice ephys experiments: If the purpose is to demonstrate that the imaging represents physiologically meaningful input from immature OSN axons onto OB neurons, it seems important to get recordings from the same part of the OB that is used for the imaging comparisons. This is especially true given that (as noted in the methods) immature OSN axons more robustly innervate the ventral OB.

In the revised slice electrophysiology experiments (see response to Reviewer 1 for details), we obtained equal numbers of recordings from the dorsal and ventral OB (see Methods, lines 891–896). This allowed us to obtain recordings from the same region as the *in vivo* imaging experiments and also get recordings from a region that is maximally innervated by immature OSN axons. There were no differences in whether dorsal vs. ventral STCs were more likely to receive input from either mature or immature OSN axons (see graph below).

4. Detection and habituation behavioral experiments look underpowered based on the variability of the post-MMZ conditions. The main claim is that immature OSNs rescue olfactory guided behavior at 5+ days post MMZ based on the lack of a significant difference between saline and MMZ at 5+ days. However, each behavior still seems substantially impaired at 5 days, but fails to reach significance because of the noisiness of the behavioral assays. For example, the claim that there is no improvement with immature OSN integration could just as easily be made from the data shown in Fig6B, F, G, and H considering that there is no significant difference between 3d post MMZ and 5, 6, or 7d post MMZ.

We have refined our protocols for the buried food and habituation-dishabituation assays (see Methods, lines 1057-1087) and repeated these assays with a new cohort of C57BL/6J mice (n = 12 per group). We now find a significant difference between the 3-day and 7-day post-MMZ groups in the buried food assay (Fig. 7B). We also find significant differences between the 3-day and 5, 6 and 7-day post MMZ groups for both habituation (Fig. 7G) and dishabituation (Fig. 7H). Hence, these new data much better support our conclusion that olfactory-guided behavior can be mediated by immature OSNs.

Importantly, we do not claim that odor detection or discrimination is fully recovered at 5-7 days post-MMZ. Our key conclusions are that *“mice can detect odors using immature OSNs alone”* (lines 524-525) and that *“odor discrimination begins to recover prior to the emergence of mature OSNs”* (lines 534-535). We have clarified this point by making statistical comparisons to the 3-day post-MMZ group for all behavioral assays, in order to determine whether performance in each assay has significantly improved at 5, 6 or 7 days post-MMZ (Fig. 7). We have also added a specific statement that performance in the odor detection and discrimination assays does not recover to control levels in the Discussion (lines 727-731): *“Notably, performance on the odor detection and discrimination assays did not fully recover by 7 days post-MMZ relative to the saline control group. This is not surprising given that saline control mice receive odor input not only from a larger number of immature OSNs but also from a high density of mature OSNs.”*

5. Considering the claims that are hinging on behavior post-MMZ (in the context of no mature OSNs) it would be useful to consider (1) imaging from immature axons after MMZ (showing activity in the OB), and/or (2) slice ephys after MMZ showing that monosynaptic connectivity is restored by immature OSNs after 5+ days

We thank the Reviewer for this suggestion and did attempt to perform both of these experiments. Unfortunately, the persistence of a high density of degenerating immature OSN axons at 5-6 days post-MMZ, which has been described previously (e.g. Fig. S2 in Tsai & Barnea Science 344 197 (2014); Fig. 4 in Kikuta et al., J. Neurosci. 36 2657 (2015); Fig. 1,2 in Blanco-Hernandez et al., PLoS One 7:e46338 (2012)) meant that neither experiment was technically feasible. Both for *in vivo* imaging in G γ 8-GCaMP6s mice and slice electrophysiology in G γ 8-ChIEF-Citrine mice, we were unable to identify what would be very sparse newly generated immature OSN axons amongst a much higher density of degenerating axons in which reporter expression was still present.

However, we were able to perform an alternative experiment that provides important new data demonstrating that the axons of newly generated immature OSNs reach the OB as early as 5 days post-MMZ. We injected biotin-labeled dextran, a retrograde tracer, into the anterior dorsal OB of mice that had received MMZ 5 days earlier. This method has been used previously to identify OSNs that project into glomeruli (Rodriguez-Gil et al., PNAS 112 5821 (2015)). We identified sparse, tracer labeled immature OSNs in the OE when mice were perfused 24 hours later (Fig. 6H). These data are described in the Results (lines 468-474) and Discussion (lines 706-709). This finding strengthens our data demonstrating that mice begin to detect and discriminate odors as early as 5 days post-MMZ, at a time point when no mature OSNs are present in the OE (Fig. 6A-G).

Minor points

1. Why in some cases is the median shown with individual values, while in other cases box and whisker plots (but no individual data) is shown? (Fig1 F vs G) Does this imply that in Fig1 G all

responding glomeruli are treated as independent? Responding glomeruli should be nested within animals to account for dependence within animals (window quality, location, overall expression, etc.).

We have revised our Figures to show data points for individual glomeruli and mean or median values for each mouse. All statistical tests compare either mean or median values per mouse, or individual glomerulus values nested by mouse, as now described in the Methods- Statistics section (lines 1069-1071). For example, the data shown in the previous Fig. 1F, which employed box and whisker plots, are now shown in Fig. 2D,E as values for individual glomeruli and mean values for each mouse. These data were analyzed with a nested one-way ANOVA for each genotype.

2. It would be useful to see example images of the staining in the MMZ conditions used to count immature and mature OSNs.

We have added examples of GAP43 and OMP staining of the septal OE in 3 – 7 days post-MMZ and saline-injected mice (Fig. 6A) used for quantification of OE width and OSN density in Fig. 6B-F. We have also added examples of OMP stained turbinates from 3 – 7 days post-MMZ and saline-injected mice (Fig. 6G).

3. Typo in line 233 (partial sentence duplicated).

This text has been replaced in the new version of the manuscript.

4. Detection behavior in which mice max out at an arbitrary time should not be analyzed as a continuous variable (i.e. in the fitting in Fig 6C). There actually does appear to be a negative correlation between digging time and time to find the buried food that is masked by all the arbitrary “10 min” data points.

We now use success in the buried food assay as a discrete variable and find no difference in digging time during the 5 min acclimation period between mice that found the buried food vs. those that failed the task (see new Fig. 7C).

5. Please show error bars in Fig 6F.

Error bars are now shown in Fig. 7F (equivalent to the previous Fig. 6F).

Reviewer 3

We would like to thank Reviewer 3 for their positive comments, noting that our study provides important new insights, is well executed, and employs multiple approaches. Specific concerns are addressed below.

In figure 1b, it is not possible to assess the potential overlap in the transition from Ggamma8 to OMP in a merged image. Although they state that there is only 3% co-labeling that is likely highly dependent on the thresholds that they use in a presumably subjective counting experiment.

We have added single channel images of GFP staining (to detect G γ 8-GCaMP6s) and OMP staining, in addition to the merged image, in Fig. 1B. When quantifying the % co-labeling, we did not threshold the images as we wanted to ensure that we would detect any OSNs with weak expression in one or both channels. To confirm that we had not missed any co-labeled OSNs with weak green (GFP) or red (OMP) fluorescence, we quantified fluorescence intensity in OSNs that we had defined as OMP+GCaMP6s+, OMP+GCaMP6s- and OMP-GCaMP6s+ (Methods lines 1012-1020, Fig. S1). We found a significant difference between both green fluorescence in OMP+GCaMP6s+ and OMP+GCaMP6s- OSNs (Fig. S1A-C), and red fluorescence in OMP+GCaMP6s+ and OMP-GCaMP6s+ OSNs (Fig. S1D-F), in all three mice. Furthermore, there was a 2.3 – 7.6-fold difference between fluorescence intensity in the dimmest co-labeled vs. the brightest non-co-labeled cell in individual channels in the three mice. Therefore, we are confident that the 3 % co-labeling that we report is an accurate measure of OMP expression in G γ 8-GCaMP6s-expressing OSNs.

In figure 1D the Ggamma8 positive glomeruli look very different than the OMP positive ones. I am not convinced that the Ggamma8-GCaMP6s positive fibers enter into glomeruli. This is critical since it is the only image that shows the OB glomeruli (other than the low mag Figure 4G that raises similar concerns) yet this forms the basis for data in several subsequent figures and conclusions.

We thank the Reviewer for pointing out that we had not included sufficient data on this very important point in the previous version of the manuscript. We now include data from G γ 8-GCaMP6s;OMP-tdT mice, demonstrating that GCaMP6s-expressing immature OSN axons do indeed enter glomeruli and respond to odorants (Fig. 1D and Results lines 102-117). We note that the very low throughput in breeding G γ 8-GCaMP6s;OMP-tdT mice precluded their use for other experiments in the paper.

We also now specifically note in the Results (lines 144-148) that innervation of glomeruli by immature OSN axons is not uniform (e.g., Kim & Greer, *J. Comp. Neurol.*, 422 297 (2000), Cheetham et al., *Nature Comms* 7 10729 (2016)) as is evident for immature OSN axons expressing GCaMP6s (see Fig. 1D for OMP-tdT-G γ 8-GCaMP6s mice as well as Fig. 2A for G γ 8-GCaMP6s mice).

The argument about the tuning of immature and mature OSNs based on their putative expression of additional receptors is not particularly informative. If one were to suggest that mature OSNs express exactly 1 OR (that determines the target glomeruli where it coalesces) and immature OSNs express two receptors (one corresponding to the cognate OR and a random one from the entire repertoire) the glomerular-level response to individual odors would look like the response profile of the common OR.

We have edited the Discussion to considerably reduce the length of the section on odor-selective responses. While we agree with the Reviewer's point (which is included in both the original and new versions of the Discussion), the finding that immature OSNs express multiple OR transcripts (Hanchate et al., *Science* 350 1251 (2015); Tan et al., *Mol. Sys. Biol.* 11 844 (2015)) has gained a lot of attention in the field; hence, we believe that it is important to include some discussion of our data in this context.

The authors describe the experiments in figure 6DE as an odor preference when it should be characterized as a food detection test. Mineral oil is usual used as a null carrier and contributes no odor that mice could choose between.

We have removed all references to odor preference and changed the description of this assay to a two-choice odor detection assay in the Methods (lines 954, 984-997), Results (lines 514-525) and Discussion (line 704). We now use the term 'investigation ratio' rather than 'preference ratio' to define odor detection ability and compare experimental groups (Fig. 7E).

The authors are acknowledged for providing impressive, quantitative data at the septum on the efficiency of their MMZ treatment. However, MMZ is highly regional in its efficacy – considerably more efficient at the septum than in lateral and ventral areas. Either a full coronal section OMP stain or at least a statement about overall efficiency is essential to properly interpret the behavior data post-MMZ.

While we do have images of whole coronal sections stained for OMP, the degenerating axon bundles in the lamina propria (as also observed by e.g., Kikuta et al., *J. Neurosci.* 36 2657 (2015), Blanco-Hernandez et al., *PLoS One* 7:e46338 (2012)) are much brighter than newly-generated mature OSNs, which are also very sparse at 7 days post-MMZ, hence we do not think that including these images would be informative. We have instead included higher magnification images of OMP-stained lateral turbinates from mice at 3-7 days post-MMZ (Fig. 6G, Results lines 464-466).

We have also added discussion of previous studies that include data on the efficiency of MMZ-mediated OSN ablation (lines 685-697). While Genter et al. (*Toxicol. Pathol.* 23 477 (1995)) found that MMZ was less effective in causing complete destruction of the OE in ventral and lateral turbinates of rats at lower doses, more recent studies in mice that quantified OSN density have shown almost complete OSN ablation, including with the same MMZ dose that we used (Tsai & Barnea, *Science* 344 197 (2014); Kikuta et al., *J. Neurosci.* 36 2657 (2015); Blanco-Hernandez et al., *PLoS One* 7:e46338 (2012)). Most importantly, we found that mice were not able to detect or discriminate odors at 3 days post-MMZ, and hence have maintained our conclusion that it is very unlikely that any residual OSNs that survived MMZ treatment could be responsible for the odor detection and/or discrimination behavior at 5-7 days post-MMZ (Discussion, lines 699-703).

REVIEWER COMMENTS

Reviewer #1 (Remarks to the Author):

The authors have added a considerable amount of new data and elegant new experiments. Many of my concerns have been addressed and the current manuscript is greatly improved. However, I still see some points to clarify and have few recommendations for improving further the paper.

Major points:

The authors claim that Ggamma8-expressing immature olfactory sensory neurons provide a functionally distinct information to the OB about odorant concentration (Figure 3). This is an important finding that sort of justifies one of the take-home messages of the paper i.e., immature neurons provide an information that is not available from mature neurons. This finding is based on the data shown in Figure 3 D-K. However, it is not clear how these data were obtained, and I am not sure the analysis is thorough enough to push forward this conclusion. Here are some questions to address and some suggestions to clarify this part of the manuscript:

- 1) It would be useful to show typical glomerular responses at increasing odorant concentrations.
- 2) In my previous comments, I asked the authors to examine the time course of responses, not only the amplitude of the change in fluorescence. The authors added analysis of time to peak and response onset latency but not of response duration. Response duration should reflect the number of spikes in the presynaptic neurons and thus may as well encode odorant concentration in mature OSNs.
- 3) Data points in Fig.3D-G show mean odorant response amplitude per mouse. How was this value calculated? In the methods it is indicated that the mean fluorescence intensity over time was extracted for each glomerulus for each trial (line 827). Then, is the mean odorant response amplitude per mouse the sum of dF/F in each glomerulus, including those that did not respond, divided by the total number of glomeruli? I guess this is how the values were calculated since some of the reported data equal zero. Wouldn't it be more correct to examine fluorescence change per glomerulus? How does including non-responding glomeruli affect the results? I also note that response amplitudes are highly variable and that large amplitude responses are more frequent in OMP-GCaMP6s mice. Is it possible that few glomeruli with large responses over influence the average, yet are not representative of responses in most glomeruli?
- 4) Another possible explanation that could explain why responses stopped to increase at high odorant concentrations is calcium probe saturation. Was this tested?
- 5) The same questions as in point 2 apply for the comparative analysis of response latency and time to peak. How was that calculated? Per glomerulus? Per field of view?

Odorant-evoked response in the M72 glomerulus

The authors compared odorant selectivity of mature and immature OSNs in the same glomerulus (M72). This is a very nice addition to the paper. The authors report that immature and mature axons projecting into the lateral M72 glomerulus responded to a similar number of odorants (line 326). However, a closer analysis of the data in Figure 4E does not support this claim. The results suggest that, in addition to 2-hydroxyacetophenone, mature OSNs projecting to the M72 glomerulus eventually responded to 3 or 4 other odorants whereas immature OSNs at most responded to only one additional odorant in the same odorant panel (which means that immature OSNs did not respond to some odorants that activated mature OSNs). Isn't that surprising? Shouldn't we expect that mature and immature neurons, if they have the same selectivity, respond to the exact same odorants? Please discuss this point.

Electrophysiology experiments (Figure 5)

New data have been acquired to demonstrate that immature OSNs provide mono-synaptic inputs to tufted cells in young (P18-25) mice. However, postsynaptic responses have been obtained in only 4 tufted cells, which is a small number. Moreover, only 3 of these 4 responses are included in

the statistical comparison with the responses mediated by mature OSNs. N=3 is not enough for a rigorous statistical comparison and may lead to aberrant conclusions. For instance, the previous report of the team as well as this study suggest, even though this is not quantified, that Ggamma8-expressing neurons are less abundant than mature OSNs in 3-4-week-old mice. Thus, one would expect that synaptic responses should be, on average, smaller in Ggamma8 mice. The authors here conclude that responses are of similar amplitudes (line 405)... If this was true, it would have important implications (more synapses per immature axons, more release site per synapse etc...). I hope additional responses will be added to this dataset. For the least, the cell that showed a response but was not included in the analysis could be included.

Reviewer #2 (Remarks to the Author):

I think the authors have done a good job revising the manuscript. They have not only addressed my main concerns (via additional data/analyses or by clarification in text), but they have done an acceptable job with other raised concerns, which were fairly extensive.

In my perspective, this manuscript should be considered for publication at Nature Communications.

Reviewer #1

We were pleased that Reviewer 1 felt that many of their concerns had been addressed by our previous revisions, that the current manuscript was greatly improved, and that we had added elegant new experiments. We have addressed Reviewer 1's additional requests for clarifications and recommendations point-by-point below.

Major points:

The authors claim that Ggamma8-expressing immature olfactory sensory neurons provide a functionally distinct information to the OB about odorant concentration (Figure 3). This is an important finding that sort of justifies one of the take-home messages of the paper i.e., immature neurons provide an information that is not available from mature neurons. This finding is based on the data shown in Figure 3 D-K. However, it is not clear how these data were obtained, and I am not sure the analysis is thorough enough to push forward this conclusion. Here are some questions to address and some suggestions to clarify this part of the manuscript:

1) It would be useful to show typical glomerular responses at increasing odorant concentrations.

We have added examples of typical responses evoked by the four tested concentrations of an odorant for glomeruli from an OMP-GCaMP6s mouse and a Gγ8-GCaMP6s mouse (Fig. 3A,B).

2) In my previous comments, I asked the authors to examine the time course of responses, not only the amplitude of the change in fluorescence. The authors added analysis of time to peak and response onset latency but not of response duration. Response duration should reflect the number of spikes in the presynaptic neurons and thus may as well encode odorant concentration in mature OSNs.

We thank the Reviewer for this suggestion and agree that the number of action potentials is likely reflected not only in the peak response amplitude, but also in the response duration. We initially considered including analysis of the decay time constant of responses but after a pilot analysis, found that it was not informative as an independent measure in terms of providing a correlation with the number of spikes in the presynaptic neurons, as response amplitude was not correlated to decay time constant (Pearson's $r = 0.20$, $P = 0.30$, $n = 30$ responses) and responses of a similar amplitude could have very different decay time constants (see graph).

The temporal resolution of GCaMP6s signals is not sufficient to permit spike deconvolution: in order to resolve individual spikes using GCaMP6s, they must be separated by an interval of 100 – 150ms (Chen et al. Nature 499 295 (2013)), whereas instantaneous odorant evoked firing rates in OSNs can exceed 50 Hz (Connelly et al., J. Neurophysiol. 110 55 (2013)). However, we reasoned that the integral of the responses (i.e., the area under the curve) could provide a useful correlate of the number of spikes and provides information that also encompasses the time course of the response.

Therefore, we now include analysis of the effect of odorant concentration on response integrals in G γ 8-GCaMP6s and OMP-GCaMP6s mice. This analysis proved informative, and we found a very similar patterns of results as for the analysis of peak DF/F amplitude. Odorant concentration had a significant effect on response integral in both OMP-GCaMP6s ($P = 0.003$) and G γ 8-GCaMP6s ($P = 0.015$) mice (mixed effects analysis of response integral per mouse). In OMP-GCaMP6s mice, comparison of mouse-odorant pairs showed that there was a significant difference in response integral between 0.5% and 1% concentrations, and between 1% and 5% concentrations, but not between 5% and 10% concentrations. In contrast, in G γ 8-GCaMP6s mice, there was a significant difference in response integral for all three ascending concentration pairs. We concluded that response integral does encode information about odorant concentration, and we have added these data to the manuscript (Fig. S4E,F; Results lines 266-278).

3) Data points in Fig.3D-G show mean odorant response amplitude per mouse. How was this value calculated? In the methods it is indicated that the mean fluorescence intensity over time was extracted for each glomerulus for each trial (line 827). Then, is the mean odorant response amplitude per mouse the sum of dF/F in each glomerulus, including those that did not respond, divided by the total number of glomeruli? I guess this is how the values were calculated since some of the reported data equal zero. Wouldn't it be more correct to examine fluorescence change per glomerulus? How does including non-responding glomeruli affect the results?

We apologize for not clearly explaining how we performed our analysis. We did not include non-responding glomeruli in the analysis of DF/F shown in Fig. 3F-I [previously Fig. 3D-G] and agree that doing so would have had a substantial impact on the results. For the response amplitude analysis, our goal was to focus on responding glomeruli, as we had separately determined the effect of odorant concentration on the percentage of glomeruli that responded to each odorant.

To calculate the mean odorant response amplitude per mouse, we included all glomeruli that responded to any concentration of a particular odorant. However, because not all glomeruli responded to all concentrations of a given odorant, there are values of zero for some mice at some concentrations. We have added additional information to the Methods (lines 942-950) to clarify this point.

We had analyzed our odorant concentration data on a per mouse basis following Reviewer 2's previous comment (on the initial manuscript) that this was the most statistically rigorous and

conservative approach to avoid biasing our analysis towards mice that either had larger numbers of glomeruli in the imaged fields of view or had more odorant responsive glomeruli. In particular, Reviewer 2 was concerned that the assumption that all glomeruli from all mice are independent is unlikely to be valid. This could be due both to potential differences in window location and quality between mice, and because we could have sampled partially overlapping subsets of glomeruli from multiple mice (e.g., M72 glomeruli from different mice would be expected to respond much more similarly to one another than would glomeruli innervated by OSNs expressing different ORs).

However, keeping these caveats in mind, we were interested also to analyze the fluorescence change per glomerulus. Indeed, we found a very similar pattern of results to our per mouse analysis (compare to Fig. 3H,I). One-way repeated measures ANOVAs with Sidak's multiple comparisons on glomerulus-odorant pairs showed that in OMP-GCaMP6s mice, there was a significant ($P < 0.001$) increase in response amplitude between 0.5% and 1% odorant concentrations, and between 1% and 5% odorant concentrations, but responses to 5% and 10% odorant concentrations were similar in amplitude ($P = 0.50$). In contrast, in G γ 8-GCaMP6s mice, there was a significant ($P < 0.001$) increase in response amplitude for each ascending pair of odorant concentrations, as we found with the mouse-odorant pair analysis. We have added these data to the manuscript (Fig. S4A,B; Results lines 252-257).

I also note that response amplitudes are highly variable and that large amplitude responses are more frequent in OMP-GCaMP6s mice. Is it possible that few glomeruli with large responses over influence the average, yet are not representative of responses in most glomeruli?

In order to address this interesting question, we repeated our analysis of the effect of odorant concentration on response amplitude using values normalized to the response amplitude for the 10% concentration of each odorant for each glomerulus. This ensured that each glomerulus contributed equally to the data set for a mouse, rather than large amplitude responses potentially skewing the mean value. We found a similar pattern of results to those in Fig. 3F-I. Mean odorant response amplitude increased with odorant concentration in both OMP-GCaMP6s mice and G γ 8-GCaMP6s mice ($P < 0.001$ for effect of concentration; two-way ANOVA). We then used repeated measures one-way ANOVAs with Sidak's multiple comparisons to compare normalized response amplitude in mouse-odorant pairs and found a similar pattern of results to those in Fig. 3H,I. In OMP-GCaMP6s mice, normalized response amplitude showed a trend to increase between 0.5% and 1% concentrations, increased significantly between 1% and 5% concentrations, and was very similar between 5% and 10% concentrations. In G γ 8-GCaMP6s mice, normalized response amplitude increased significantly between 0.5% and 1% concentrations, showed a trend to increase between 1% and 5% concentrations, and increased significantly between 5% and 10% concentrations. Therefore, it does not appear that a small number of strongly responding glomeruli in OMP-GCaMP6s mice underlies the different relationships between odorant concentration and response amplitude in OMP-GCaMP6s vs. G γ 8-GCaMP6s mice. We have added these data to the manuscript (Fig. S4C,D; Results lines 257-264).

4) Another possible explanation that could explain why responses stopped to increase at high odorant concentrations is calcium probe saturation. Was this tested?

This is an important point, especially in light of the differences in concentration-dependent responses that we report between OMP-GCaMP6s and G γ 8-GCaMP6s mice, which we have now addressed in the manuscript. During our original data analysis, we had visually inspected all fluorescence traces for evidence of GCaMP6s saturation at higher odorant concentrations (now included in the Methods, lines 926-929), and found no evidence of responses containing plateaus. We have also now performed additional analysis of the largest glomerular odorant responses (those with DF/F values greater than 1000 %) in OMP-GCaMP6s mice to confirm that they did not show evidence of GCaMP6s saturation (Methods lines 929-940; Results lines 245-246; Fig. S3). For each glomerulus that exhibited a maximal DF/F value >1000 %, we compared the maximal responses to a smaller amplitude response evoked by the same odorant. Specifically, we calculated the number of frames between the maximum rising phase of the response and the first non-positive change in fluorescence. We reasoned that if GCaMP6s saturation was occurring, responses would rapidly (within 1-2 frames) transition from the rising phase to a flat plateau. Furthermore, saturation would not be expected to occur in the smaller amplitude responses from the same glomeruli; hence, by comparing the number of frames between the maximum rising phase of the response and the first non-positive change in fluorescence for maximal vs. smaller amplitude responses for individual glomeruli, we could perform an additional test for saturation. We found no evidence of responses with very rapid transitions between the rising phase of the response and the first non-positive value, and also found no significant difference in this parameter between maximal vs. smaller amplitude responses for the same glomeruli. Therefore, we concluded that GCaMP6s saturation was very unlikely to contribute to the differences in concentration-dependent responses that we found between OMP-GCaMP6s and G γ 8-GCaMP6s mice.

5) The same questions as in point 2 apply for the comparative analysis of response latency and time to peak. How was that calculated? Per glomerulus? Per field of view?

We again apologize for not clearly explaining how these data were calculated in the previous version of the manuscript. The data presented in Fig. 3J-M showing differences in response latency and time to peak were also calculated on a per mouse basis, for responding glomeruli only. We have now added additional information to the Methods (lines 945-950) to clarify these analyses.

Odorant-evoked response in the M72 glomerulus

The authors compared odorant selectivity of mature and immature OSNs in the same glomerulus (M72). This is a very nice addition to the paper. The authors report that immature and mature axons projecting into the lateral M72 glomerulus responded to a similar number of odorants (line 326). However, a closer analysis of the data in Figure 4E does not support this claim. The results suggest that, in addition to 2-hydroxyacetophenone, mature OSNs projecting to the M72 glomerulus eventually responded to 3 or 4 other odorants whereas immature OSNs at most responded to only one additional odorant in the same odorant panel (which means

that immature OSNs did not respond to some odorants that activated mature OSNs). Isn't that surprising? Shouldn't we expect that mature and immature neurons, if they have the same selectivity, respond to the exact same odorants? Please discuss this point.

We thank the Reviewer for this suggestion and have edited the Results (lines 382 - 384) and Discussion (lines 721-725) to better describe the data shown in Fig. 4E, as highlighted by the Reviewer. We have also added further discussion of possible explanations of the number of odorants to which lateral M72 glomeruli responded (lines 725-732).

Electrophysiology experiments (Figure 5)

New data have been acquired to demonstrate that immature OSNs provide mono-synaptic inputs to tufted cells in young (P18-25) mice. However, postsynaptic responses have been obtained in only 4 tufted cells, which is a small number. Moreover, only 3 of these 4 responses are included in the statistical comparison with the responses mediated by mature OSNs. N=3 is not enough for a rigorous statistical comparison and may lead to aberrant conclusions. For instance, the previous report of the team as well as this study suggest, even though this is not quantified, that Ggamma8-expressing neurons are less abundant than mature OSNs in 3-4-week-old mice. Thus, one would expect that synaptic responses should be, on average, smaller in Ggamma8 mice. The authors here conclude that responses are of similar amplitudes (line 405)... If this was true, it would have important implications (more synapses per immature axons, more release site per synapse etc...). I hope additional responses will be added to this dataset. For the least, the cell that showed a response but was not included in the analysis could be included.

Unfortunately, we were not able to add additional recordings to the dataset, due to the data acquisition system that we used no longer being available. However, as requested, we have added data for the previously excluded cell that showed a response to the manuscript. This cell had previously been omitted because although it showed monosynaptic responses to a 1 ms 50 % intensity stimulus, it had an incomplete dataset and lacked sweeps for the 1 ms 100 % intensity stimulus that we had used for our analysis in Fig. 5 of the previous version of the manuscript. To avoid combining data collected with different stimulus parameters, we have now added Figure S6, which shows data collected from all STCs for which we collected sweeps using a 1 ms 50 % intensity stimulus. These data comprise: 6 of the 7 STCs from OMP-ChIEF-Citrine mice shown in Fig. 5; and 2 of the 3 STCs shown in Fig. 5 plus the 1 additional STC (from a different mouse) that was omitted from Fig. 5 from Gγ8-ChIEF-Citrine mice. The two cells (one from an OMP-ChIEF-Citrine mouse and one from a Gγ8-ChIEF-Citrine mouse) for which data are shown in Fig. 5 but not Fig. S6 lacked sweeps for the 1 ms 50 % intensity stimulus. Overall, we now present monosynaptic response data from 7 STCs from 5 OMP-ChIEF-Citrine mice, and 4 STCs from 3 Gγ8-ChIEF-Citrine mice.

We have added discussion of the interpretation of the similar STC response amplitudes in Gγ8-ChIEF-Citrine and OMP-ChIEF-Citrine mice (lines 645-654) and also edited the Results (lines 470-472) and Discussion (line 656) to highlight that our data should be interpreted with caution due to the small number of STCs analyzed. We have also made minor edits to the Fig. 5 legend (lines

499-502 and lines 1375-1378) and added additional information to the Methods (lines 1030-1057) to clarify what data were included in Fig. 5 and Fig. S6.

Reviewer #2

We were pleased that Reviewer 2 felt that we had both addressed their main concerns and done an acceptable job in addressing concerns raised by other Reviewers. Reviewer 2 did not request any additional revisions.

REVIEWERS' COMMENTS

Reviewer #1 (Remarks to the Author):

The authors have addressed my concerns with additional analysis and clarification in the text. I recommend publication.

Response to Reviewers

REVIEWERS' COMMENTS

Reviewer #1 (Remarks to the Author):

The authors have addressed my concerns with additional analysis and clarification in the text. I recommend publication.

Reviewer 1 did not request any additional revisions.